

# A New Versatile Dropsonde for Atmospheric Soundings with HALO – The KITsonde

Christoph Kottmeier[1], Andreas Wieser[1], Ulrich Corsmeier[1], Norbert Kalthoff[1], Philipp Gasch[1], Bastian Kirsch[1], Dörthe Ebert[2], Zbigniew Ulanowski[3], Dieter Schell[4], Harald Franke[4], FlorianSchmidmer[5], Johannes Frielingsdorf[5], Thomas Feuerle[6], Rudolf Hankers[6]

[1]Institute of Meteorology and Climate Research Troposphere Research (IMKTRO), Karlsruhe Institute of Technology (KIT), Karlsruhe, Germany

[2]German Weather Service, Offenbach, Germany

[3]British Antarctic Survey, Cambridge, United Kingdom and University of Manchester, Manchester, United Kingdom

[4]enviscope GmbH, Frankfurt am Main, Germany

[5]Graw Radiosondes GmbH & Co KG, Nürnberg, Germany

[6]Technische Universität Braunschweig, Braunschweig, Germany

*Correspondence to*: Christoph Kottmeier (Christoph.Kottmeier@kit.edu)

**Abstract.** A new modular multi-sensor aerological dropsonde system for high and fast-flying research aircraft has been developed for studying atmospheric processes. This new system allows to drop release containers with up to 4 sondes inside and data from up to 30 sondes can be transmitted simultaneously. After separation from the release container, the sondes enable high-resolution spatio-temporal profiling of temperature, humidity, pressure, and wind with a time resolution of 1.2 s corresponding to approximately 10 m vertical resolution. The modular design ensures simple integration of additional sensors without extensive flight tests and recertification for e.g. particle measurements and radioactivity.

The standard meteorological sonde comprises sensor elements of a commercial Graw DFM-17 radiosonde, a 400 to 406 MHz band communication link to the aircraft, and an optional satellite communication module. By means of the satellite link, the data can be made available worldwide in near real time and data loss is avoided when the dropping aircraft leaves the telemetry range.

The main feature of the new system is the release container, which allows for dropping through standard dropsonde dispensers of both mid-size turbo-prop aircraft (e.g. Dornier Do 128-6) and jet aircraft (e.g. the Gulfstream 550 "High Altitude and Long Range Research Aircraft, HALO"). The release container ensures safe separation from the aircraft and protects its payload during deceleration from aircraft speed to fall speed before the sondes are released by an electro-mechanical mechanism. Operations in different campaigns have confirmed the reliability of the entire system and the quality of acquired data. Feasibility of the technical and operational approach for targeted observations of a mesoscale convective system in Argentina



was demonstrated by HALO measurements during the SouthTRAC (TRAnsport and Composition of the southern hemisphere UTLS campaign) campaign. Moreover, a configuration consisting of one meteorological sonde coupled (a) with an optical counter for particle sizing was tested using a Dornier Do 128-6 aircraft during a Saharan dust episode over Germany and (b)
with a radioactivity sensor was successfully dropped from a Learjet 35A.

## 1    Introduction

Aerological observations of pressure, temperature, humidity, and wind in the atmosphere are indispensable for weather
forecasting and atmospheric research (Vömel and Fujiwara, 2021). The first parachute radiosondes were used in the 1940s. Compared to radiosonde or tethersonde systems, dropsondes are operated for targeted observations often over oceans or remote land or ice surfaces that can hardly be accessed by other in-situ measurement systems. Their spectrum of operation in previous studies covers a large range of different spatio-temporal scales and atmospheric phenomena, as outlined below.

Dropsondes made by different manufacturers are being operated in large numbers by national weather services (NWS) and
research institutions. Early attempts to use dropsondes with the capability to measure wind based on the OMEGA navigation system were made in the 1970s, followed by systems based on LORAN-C and OMEGA in the 1980s. In 1994, NCAR, German Aerospace Center (DLR), and NOAA agreed to develop a GPS dropsonde (RD93) based on Vaisala radiosonde technology (Cole, 1997; Wick et al., 2018). The dropsonde with a four-channel telemetry system was designed for release from a jet aircraft flying at tropopause level, with the telemetry system allowing for a horizontal spacing of 100 km between drops. In
the following decades, NCAR developed several generations of dropsondes and contracted Vaisala Inc. to manufacture them in considerable number. Currently, Vaisala is producing several thousand dropsondes per year for worldwide use. Most frequent applications are for NWS hurricane surveillance & reconnaissance, hurricane research, atmospheric rivers research, as well as naval research and ocean monitoring, with a regional focus on the western Atlantic and central Pacific. Wang et al. (2018) also presented a dropsonde for use on unmanned vehicles. A system with re-usable dropsondes, without telemetry
bandwidth limiting the number of simultaneous sondes, was developed and applied by Kottmeier et al. (2001). To measure additional data and to avoid frequency limitations, measured data were no longer transmitted via radio link, but stored internally. Landing position data were transmitted via mobile phone communication. The sondes were recovered after operation for data recovery and repeated use. Meisei (2018) in Japan and Yankee in the USA (Black et al., 1996) also manufacture and distribute dropsondes, recently joined by SkyFora (2024). For more details about available dropsonde
systems, we refer to Vömel and Fujiwara (2021).

There are numerous examples of research results obtained by using data from dropsondes in the last decades. Densely spaced drop soundings were made in an Arctic research program (Hartmann et al., 1996) to study the evolution of cold air outbreaks over the Northern Atlantic Ocean in the marginal sea ice zone. Dropsonde data also helped to investigate structures and phenomena of the atmospheric boundary layer (e.g. Ralph et al., 2005; Flamant et al., 2007; Wang et al., 2008; Messager et
al., 2010; Zhang et al., 2013; Schmidt, 2014). Barthlott et al. (2006) used dropsondes to investigate mesoscale circulation





relevant to triggering convection, Groenemeijer et al. (2009) used dropsonde data to analyse the kinematic and thermodynamic conditions in the vicinity of convective storms. Szunyogh et al. (2000), Kim et al. (2010), Zhang et al. (2013), Wang et al. (2015), Young et al. (2015), and Kren et al. (2016) used dropsonde data to investigate typhoons, storms, hurricanes, and cyclones. Area averaged vertical motion was estimated from divergence calculation using multiple dropsondes (Bony and
Stevens, 2019). More recently, dropsondes deployed during the EUREC[4]A (Elucidating the role of clouds-circulation coupling in climate) field campaign helped to characterise mesoscale circulations over the tropical North Atlantic (George et al., 2021). Finally, Burpee et al. (1996), Weissmann et al. (2011), Jung et al. (2012), and Schindler et al. (2020) used dropsonde data in a combination of data assimilation and weather forecast. Dropsondes have also been used for aerosol characterisation and trace gas profiles (Vömel et al., 2023).

This publication introduces the new "KITsonde" system, a dropsonde for atmospheric research, which was developed for several reasons. (1) It shall be able to receive signals from up to 30 meteorological sondes at the same time in order to reach a much denser spatio-temporal density of measurements. (2) The use of state-of-the-art sensors and telecommunication system shall allow for an easy adaptation to new developments in sonde technology. (3) Besides the dynamic and thermodynamic states of the atmosphere, system extensions shall provide additional information covering particle size and shape from optical
measurements, radioactivity, or chemical constituents of the atmosphere. (4) A release container for dropsondes which can be operated on board of all research aircraft using standard dispensers shall allow for a great flexibility in operations. To fulfil these objectives, the novel KITsonde was developed by KIT and its partner institutions as a cost-efficient dropsonde system. The paper introducing this system is structured as follows. Section 2 describes the technical details of the meteorological sonde, of the sensors for particles and radioactivity, and of the release container and cabin installations. Section 3 provides exemplary
measurements with the KITsonde deployed during the "TRAnsport and Composition of the southern hemisphere UTLS campaign (SouthTRAC)" onboard the High Altitude and Long Range Research Aircraft (HALO). They demonstrate the feasibility of the new technical and operational approach for targeted observations of a mesoscale convective system. Tests of the system coupled with an optical particle counter, the Universal Cloud and Aerosol Sounding System (UCASS) are also described. During some of the UCASS tests, shallow clouds as well as a Saharan dust layer transported over Europe were
sampled and characterised.

## 2 KITsonde's innovative features, technical description, and design

A schematic overview of the KITsonde system is shown in Fig. 1. The innovative feature of the KITsonde system is its possibility to operate up to 30 sondes simultaneously, which can be dropped in batches of up to four at short intervals. A newly developed release container can carry up to four small and lightweight meteorological sondes during release from research
aircraft, that are equipped for the Advanced Vertical Atmospheric Profiling System (AVAPS) with a standard RD93/94/41 dispenser. The release container protects the sondes from the strong mechanical forces during launch by delayed release of the sonde. As the sondes contained in one release container are equipped with individually sized parachutes, they can be separated from each other when falling. The container inserts are made of sintered polyamide, which allows for a rapid and cost-effective





adaptation for the integration of new sensors and interfaces with minimal effort. Extensive flight testing and recertification can
be avoided for modifications within an envelope concept for total weight and centre of gravity. As a first extension, the
meteorological sonde was combined with a newly developed probe measuring particulate matter and cloud droplets (Sect. 2.5).
This extension was developed, integrated, and successfully tested in collaboration with the University of Hertfordshire (Smith
et al., 2019). Dropsondes using the new release container and measuring radioactivity with a new gamma dose rate sensor in
cooperation with the German Weather Service (DWD) are currently under development (Sect. 2.6).

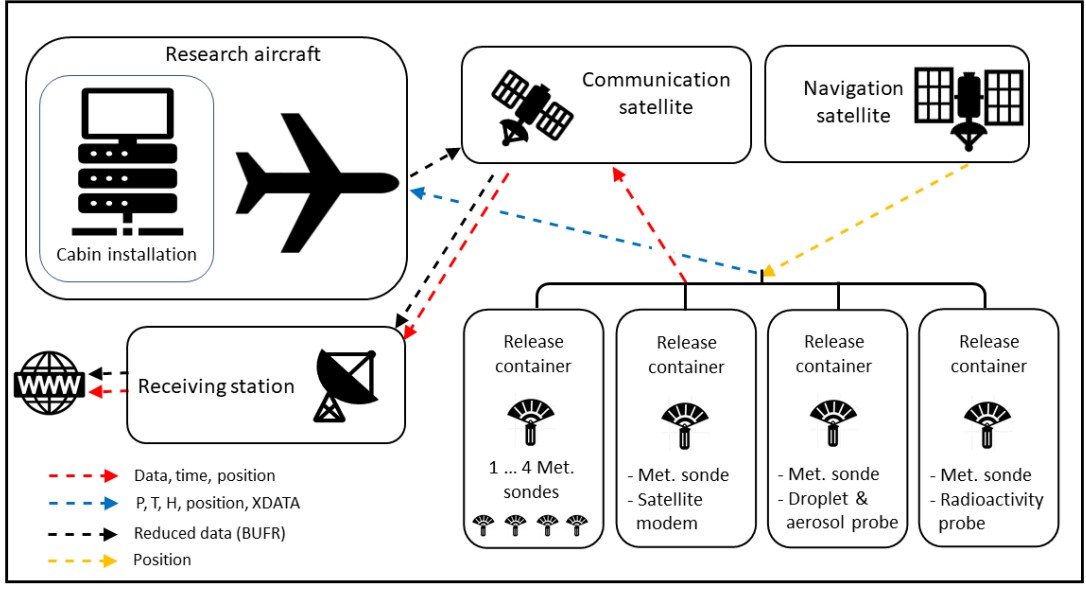


**Figure 1: Schematic overview of the KITsonde system.**

The data of the dropsondes are received aboard the research aircraft using the 400 to 406 MHz band direct telemetry.
Optionally, a direct satellite link from the dropsonde can transfer data to the Earth's surface (Fig. 1). For that purpose, an
additional satellite communication module (satcom) is connected to a meteorological sonde and dropped with the release
container. The microcontroller on the module collects the data from the sonde, provides for the satellite connection, and
transfers the data to a server on ground. With the satcom module, the sounding data are available on the internet in near real
time which is advantageous for data assimilation and mission coordination during large field campaigns. Iridium
communication works even when the sondes are launched at remote locations and the launching aircraft is out of the range of
the direct telemetry link to the aircraft. These features give the KITsonde an unprecedented flexibility with respect to detection
of atmospheric parameters and resolution in time and space. If a satcom module is used in the release container, there is free
space for one meteorological sonde only.



The KITsonde system consists of three main components (Fig. 2): firstly, the cabin installations for sonde initialisation, sonde reception after dropping, and data receipt, decoding, and storage, secondly, the multipurpose release container, and lastly the individual sondes for meteorological, particle, and radioactivity sensors contained in the release container. The different

components of the KITsonde system are introduced in some detail in the following section.

## 2.1 Cabin installation with data acquisition and data transfer to aircraft

Besides the AVAPS RD93 compatible dispenser system, cabin installations for the KITsonde system include five units mounted in aircraft racks (Fig. 2): receiver; data acquisition; system control and near real-time data decoding unit; sonde configuration box; and keyboard, monitor, and mouse unit for operators. Configuration and near real-time data visualisation

are provided by a graphical user interface accessible in the aircraft rack (main unit) or via a network on any computer aboard the aircraft. The design of the receiver unit allows for the use of AVAPS along with the KITsonde by an amplified loop-through antenna link.

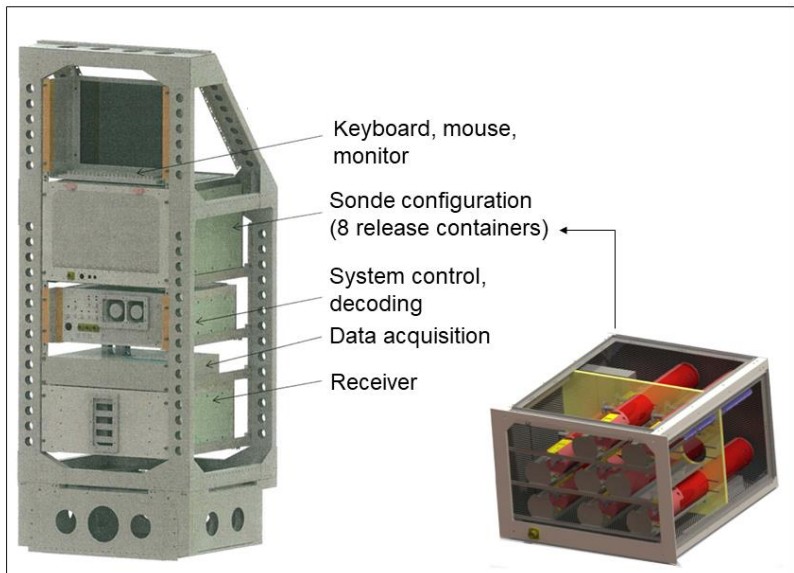

**Figure 2: KITsonde cabin installation in the aircraft.**

### 2.1.1 Sonde configuration unit

In the configuration unit the sondes packed into release containers are prepared for launch, i.e., for configuration, initialisation, and power supply until dropping (Fig. 2). The configuration unit has eight release container slots for simultaneous configuration. One dedicated receiver for each sonde is connected to a 400 MHz antenna inside the box for sonde check and a GPS repeater supplies the GPS signal. To ensure that there are no interferences with the electronics of the aircraft, the

configurations box is tested on electromagnetic compatibility according to airworthiness standards.





### 2.1.2 Receiver unit

The aircraft receiver unit (Fig. 2) is based on a traditional superheterodyne low-power consumption design. The receiver unit includes 32 receivers to receive the signals of up to 30 sondes. The receivers allow for a free choice of frequency and are equipped with automatic frequency control (AFC). Receiver 31 constantly scans the frequency band for free frequencies and

receiver 32 is used for the sonde check in the initialisation box before launch. After the initialisation of a sonde by the configuration computer, the receiver is set to the crystal-stabilised sonde frequency and data reception and the signal of the sonde is checked. The 400 MHz antenna on the lower fuselage of the aircraft receives the signals of the falling sondes and transmits them to the 30 receivers by an antenna pre-amplifier and a power splitter.

### 2.1.3 Data acquisition unit; system control and decoding unit

Three computers form the system control computer network. The configuration PC initialises the sondes by setting individual transmitting frequencies within the 400.0 MHz to 406.0 MHz band for each sonde and assign one of the receivers to the sonde. The selection of a frequency is supported by an automatic frequency scan helping to avoid local radiosonde frequencies. The frequencies assigned are typically separated by at least 25 kHz from each other. During the initialisation phase a GPS signal is provided by a GPS repeater and the data of the meteorological sonde are checked.

The calibration data of the sonde together with its serial number are stored. After the release the data are forwarded to the decoding PC where the data are digitised and transmitted to the evaluation PC. The data are set up with the calibration data and are transformed to physical parameters. A graphical interface enables the visualisation of the vertical profiles in near real time.

### 2.2 Release container

The main advantage of a release container (Fig. 3) is the safe separation from the aircraft by delayed parachute opening and the protection of the payload inside while dropping. The typical speed of operation of the HALO aircraft (Gulfstream G550) is about 250 m s$^{-1}$. Unprotected dropping of the sondes may cause severe damage to them. The dimension of the release container equals that of the Vaisala RD93/RD94. Therefore, it is compatible with the wide-spread AVAPS release systems and makes the KITsonde easily adaptable to a variety of research aircraft.




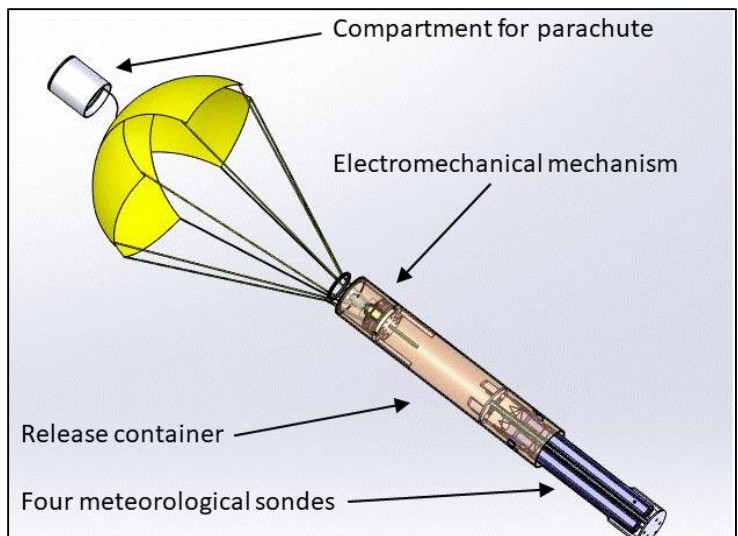

**Figure 3: Schematic drawing of the release container with parachute (yellow) and four meteorological sondes (blue). Status shown approximately 10 seconds after dropping when the parachute opens and during release of four meteorological sondes.**

The dropping process is as follows (Figs. 3 and 4): immediately after release, the release container is exposed to high

deceleration and spin rates which might damage its payload (e.g. four meteorological sondes). Therefore, an electromechanical mechanism (Fig. 3) was developed to delay the release of the payload from the container for a programmable number of seconds after maximum deceleration and spin. The upper part of the release container consists of a compartment for the main parachute and its unfolding mechanism. A microprocessor unit with acceleration and rotation sensors detects 3-dimensional accelerations and spin rates 52 times per second. If both exceed a programmed level, the release container drop is detected.

After a programmed time delay, e.g. 5 seconds after the most turbulent container motion is reduced, a motor inside the release container turns a thread rod, which opens the upper closure head of the container.  Forced by air motion, the main parachute is set free. It opens, reduces container motion, and positions the complete container in vertical direction. Meanwhile, the motor still turns the thread rod and finally releases the insert holding the payload (sondes) in the cargo bay of the release container. In the case of four meteorological sondes (Fig.4b), these are now set free. Their individual parachutes open and the sondes are

falling while sampling data with a typical vertical speed between 3 m s$^{-1}$ and 10 m s$^{-1}$ depending on height and chosen parachute size.



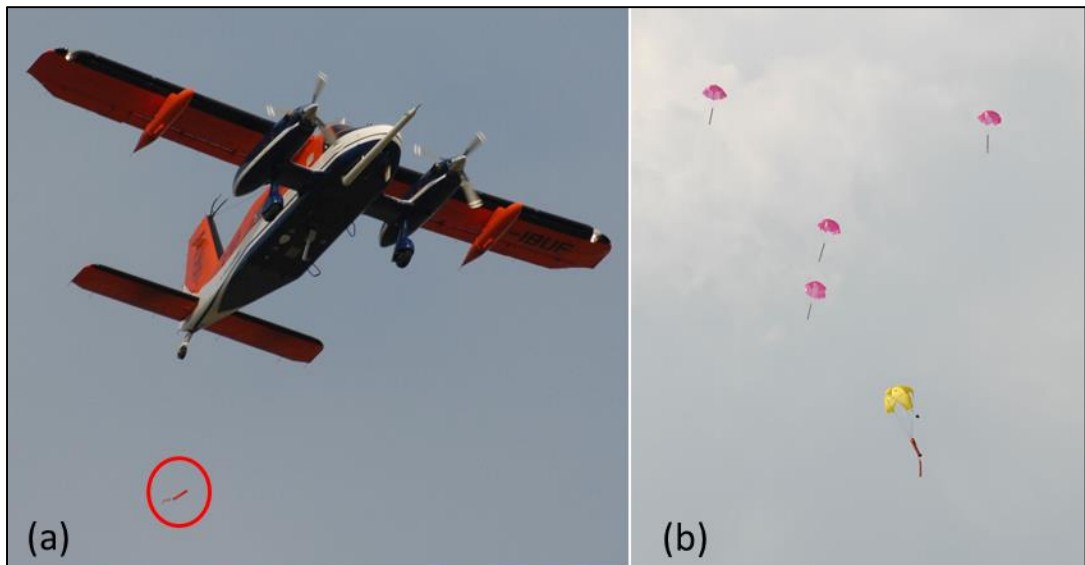

**Figure 4: (a) Research aircraft Dornier Do 128-6 dropping a release container. The red circle marks the release container shortly after dropping. (b) Release container (yellow parachute) with four meteorological sondes with red parachutes of different sizes.**

Launch identification by detection of thresholds of acceleration and rotation rate prevents unwanted releases of sondes from the container during handling and transport of activated sondes especially inside the aircraft. For operations on different aircraft, the threshold values can be defined and adjusted with the ranges of $\pm$ 78 m s$^{-2}$ for acceleration, $\pm$ 100 deg s$^{-1}$ for rotation, and 0 to 40 s for the delay time.

In order to guarantee high flexibility in operating new sensor configurations or new sensors within the release container, an "envelope concept" was established with air traffic authorities for certification of release container inserts. The flights for the certification of the release container inserts were performed with the Dornier Do 128-6 based on a preliminary approval for this aircraft. Currently, the release container is certified for releases from the Gulfstream G550 aircraft for 561 to 999 g total weight at release and a centre of gravity between 159 and 245 mm from its bottom independently of the payload, as long as it fits into this "envelope". The release container itself has a weight of 483 g and therefore allows for a total payload (sensors) ranging from 78 to 516 g. The weight of an individually designed and 3D printed insert, including sondes for meteorological and other purposes, is allowed to be within these weight limits. The centre of gravity of the packed release container can be modified to be within the envelope limits by balance weights at different positions in the release container.

### 2.3 Meteorological sonde

The active payload of the KITsonde is a meteorological sonde manufactured by Graw Radiosondes. This electronic module named EL-18 (Fig. 5a) contains the same circuitry as the regular Graw DFM-17 radiosonde, but is physically optimised for use with a dropsonde application. The DFM-17 is in widespread synoptic use by meteorological agencies worldwide. The



temperature measurement is based on a bead thermistor, the measurement range being from -90 °C to + 60 °C. Relative humidity is measured with a capacitive thin-film polymer sensor within a range of 0 to 100 %, while pressure is measured using both the GNSS module as well as a barometric pressure sensor based on a micro electro-mechanical piezo-resistive sensor element with a combined range of 1100 to 1 hPa. The state-of-the-art sensors are located on a sensor boom (Fig. 5b), which is coated with aluminium to minimise the impact of radiation on the measurement. A hydrophobic coating helps to minimise the impact of evaporative cooling (Graw, 2024).

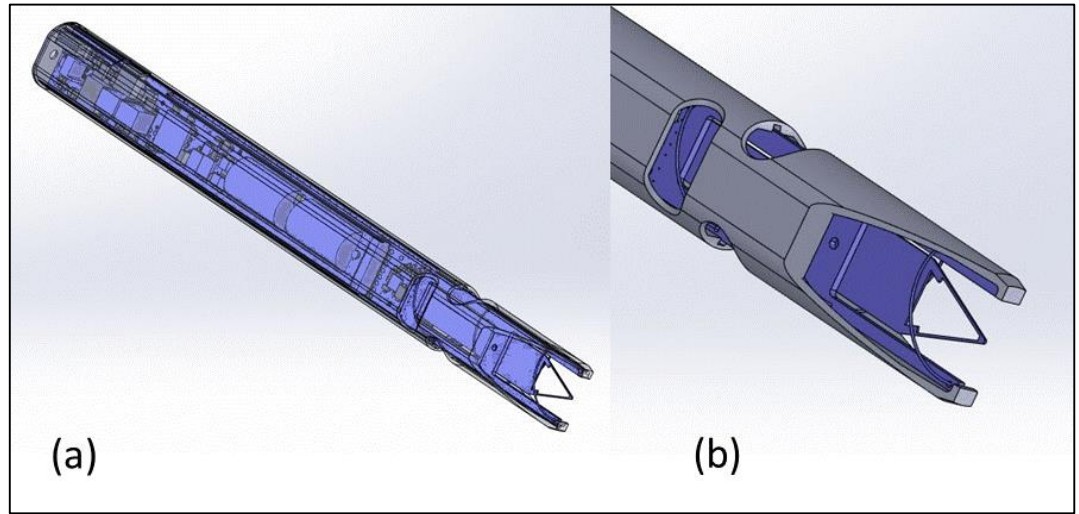

**Figure 5: (a) Schematic drawings of the meteorological sonde EL-18 and (b) sensor boom with temperature and humidity sensors.**

For wind and position retrieval, the GNSS receiver u-blox MAX-M8C is used. It can receive various combinations of the L1 signals from GPS, GLONASS, and Beidou. The cold-start time (time to first fix) is 26 s under good conditions (open sky). On board the aircraft, a GNSS signal is provided by means of a GPS re-radiation antenna located in the configuration box with a re-acquisition time of 1 s in case of a signal loss. Further specifications of the EL-18 are given in Tab. 1.

A complete set of temperature, relative humidity, and pressure (PTU) values is transmitted every 1.12 s using the transmitter of the sonde. Additionally, a GNSS dataset is transmitted every second. The modification of the dimensions compared to the off-the-shelf DFM-17 radiosonde allows the integration of EL-18 in a cylindric housing of 309 mm in length and 24 mm in diameter. Hence, the release container, which is 404 mm long and 70 mm wide, accommodates four meteorological sondes of the type EL-18 together with four parachutes, (Fig. 3).





**Table 1: Technical specifications of the electronic module of the Graw dropsonde EL-18.**

| | | |
|---|---|---|
| **Temperature** | Type | Glass-encapsulated NTC bead thermistor |
| | Measurement range | -90 °C to +60 °C |
| | Resolution | 0.01 °C (internal) |
| | Repeatability in calibration | < 0.05 °C |
| **Humidity** | Manufacturer | E+E Electronic, HMC03M |
| | Measurement range | 0 to 100 %RH |
| | Resolution | 0.1 %RH |
| | Repeatability in calibration | < 1 %RH |
| **Pressure** | Manufacturer, Type | Bosch Sensortec, BMP388 u-blox, MAX-M8C |
| | Measurement range | 1100 to 1 hPa |
| | Resolution | 0.01 hPa (internal) |
| | Uncertainty > 100 hPa | < 1 hPa |
| | Uncertainty 100 to 10 hPa | < 0.2 hPa |
| | Uncertainty 10 hPa | < 0.04 hPa |
| **Geopotential height** | Type | GPS-module u-blox, MAX-MBC- |
| | Measurement range | -500 to +40,000 m |
| | Resolution | 0.1 m |
| | Uncertainty | < 8 m |
| | Reproducibility in sounding | < 5 m |
| **Wind speed** | Type | GPS-module Tellit J-N3 |
| | Measurement range | 0 to 200 m s$^{-1}$ |
| | Resolution | 0.01 m s$^{-1}$ (internal) |
| | Uncertainty | < 0.1 m s$^{-1}$ |
| **Wind direction** | Type | GPS-module Tellit J-N3 |
| | Measurement range | 0 to 360° |
| | Resolution | 0.01° |
| | Uncertainty | < 1° |
| | Transmission interval | 1.12 s (PTU), 1 s (wind) |
| | Tuning range | 400 to 405.99 MHz |



| **Telemetry** | Bandwidth | < 12 kHz |
|---|---|---|
| | Max. range | > 250 km |
| | Frequency stability, 90 % probability | < 1 kHz |

As described above, EL-18 is directly derived from the DFM-17 radiosonde, which has been subject to several quality evaluations and intercomparisons. In particular, the sensors and analogue to digital converter circuity as well as the
manufacturing and calibration processes are the same for both sondes. The EL-18 uses a slightly modified sensor boom shape to accommodate the dropsonde application (Fig. 5b). Due to the different nature of the applications, not all corrections used in the DFM-17 data product can be used for the EL-18. In the WMO intercomparison campaign UAII 2022 (Dirksen et al., 2024), DFM-17 yielded respectable results and was found fit-for-purpose in almost all application areas with respect to the breakthrough criterion, which results in a significant improvement for the targeted application. The threshold (minimum)
requirements for temperature, humidity, and wind concerning (i) aeronautical meteorology, (ii) nowcasting, (iii) very short-range forecasting, (iv) numerical weather prediction, and (v) real-time monitoring in the whole troposphere are fulfilled by the DFM-17. DFM-17 is one of only three radiosondes, the humidity measurement of which is found to be fit-for-purpose in the free troposphere regarding the breakthrough criterion in the global numerical weather prediction and real-time monitoring applications (Frielingsdorf et al., 2024). DFM-17 (as one of six out of ten) and EL-18 are both capable of measuring relative
humidity above 100 %RH, which is important for cloud thickness detection. As for most radiosonde algorithms (Dirksen et al. 2024), an active clipping of measured relative humidity values > 100 %RH is performed. It must be noted, however, that the EL-18 does not use the heated humidity sensor of the latest DFM-17 generation, although this feature is less relevant to dropsondes than to radiosondes.

## 2.4 The satellite modem

The data of the KITsonde can also be made available worldwide in near real time through a direct sonde link to a satellite. The Iridium satellite network was chosen for the KITsonde system because of its worldwide coverage, the relatively small modems, and the availability of pre-paid SIM cards at reasonable prices. With a special service, it is possible to connect the meteorological sonde to a computer using a terminal emulation program.

The complete data transmission electronics consists of two circuit boards, an Iridium antenna, and two batteries. One of the
two circuit boards is the modem, which sends the data via satellite to the ground station. The other circuit board is a specially adapted printed electronic circuit. It consists of two microcontroller boards for controlling and a micro SD card for buffering the data. The first microcontroller board is connected via ribbon cable and converts the data of the sonde into ASCII format. The second microcontroller board writes the ASCII files on a micro SD card. Data are read from the micro SD card and sent via satellite to a second modem on the ground. This modem is connected to a computer and the data can be read and saved
with a terminal program. Two lithium batteries supply the electronic system with power. The modem sends the data via the



Iridium antenna. To avoid interferences, the antenna needs to have a high power and be placed as far as possible away from the GPS antenna. The reason is that they have a similar frequency. This buffering system is necessary to avoid data loss when the connection to the satellite is interrupted.

### 2.5 The Universal Cloud and Aerosol Sounding System (UCASS)

One sensor system which has already been interfaced to the KITsonde is the Universal Cloud and Aerosol Sounding System (UCASS, Smith et al., 2018). The system is a low-cost, miniaturised optical particle counter developed by the University of Hertfordshire/United Kingdom for the measurement of micron-scale particles, i.e. for aerosol/droplet concentrations and size distribution measurements. The size and cost of the instrument are kept to a minimum. The UCASS has up to 16 configurable size bins capable of sizing particles in typical approximate ranges of 1-20 μm and 2- 40 μm in terms of lower bin threshold of

particle diameter. While the former range is intended for coarse aerosol detection, the latter is designed for cloud droplets. Size bin boundaries are individually programmable and can be shifted together by changing electronic gain. The optical particle counter (OPC) is laboratory-calibrated to account for variables, such as laser power and optical component parameters and alignment. To reduce weight, power requirements, and cost, the use of air pumps has been avoided. Instead, UCASS relies on an "open" geometry, where air flow is provided by the movement of the OPC itself. To that end, the body of the OPC has a

wide, central channel running along its entire longitudinal axis (Fig. 6a). Particle detection takes place when particles cross a laser beam running across the channel, with the sample volume being defined optically (strictly speaking, it is a sample cross-section, with the third dimension being provided by the product of air velocity and sampling time). The shape of the OPC body is such that the air velocity at the sample volume is similar to that outside (i.e. the fall speed of the dropsonde), which ensures relatively undisturbed, nearly isokinetic sampling. This is especially important for larger particles, which can be lost or

miscounted in artificially aspirated (pumped) sampling systems (Smith et al., 2018; Girdwood et al., 2020).

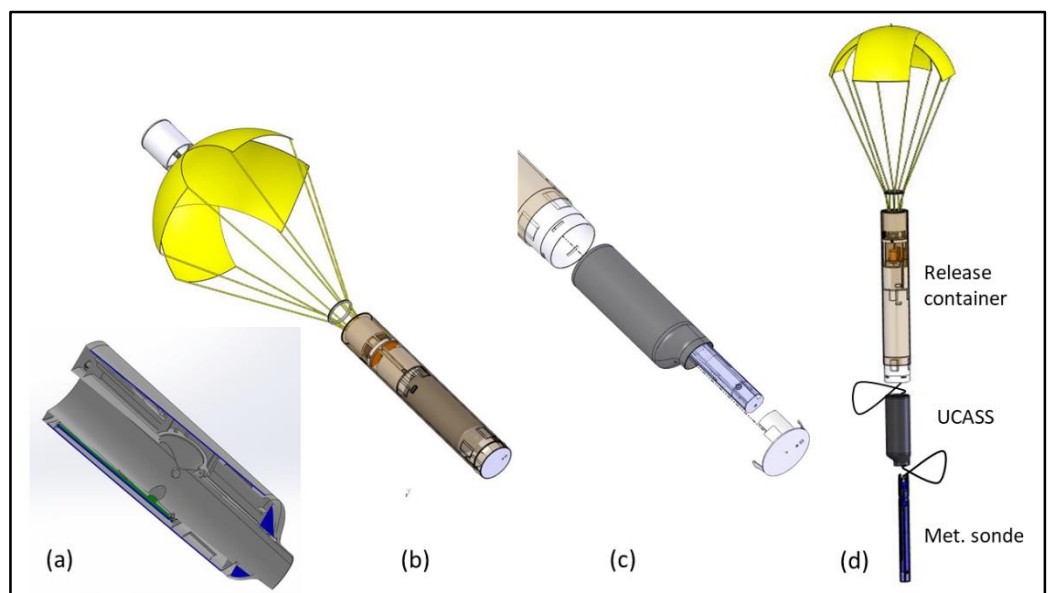



**Figure 6: (a) Cross-section of OPC UCASS, (b) sonde configuration right after dropping with open parachute, (c) UCASS set free from release container with meteorological sonde (blue) partly within UCASS, and (d) final sonde configuration during measuring flight.**

270 The UCASS is designed with this particular shape for use as a dropsonde system. The entire OPC body fits into a standard KITsonde release container (Fig. 6b). The UCASS is 180 mm long, 64 mm in diameter, and weighs 0.28 kg. While in the container, the connected meteorological sonde fits into the central channel of the UCASS (Fig. 6c). Particles are counted in individual size bins in parallel, normally over periods of 1 s. When UCASS is used as a stand-alone instrument, the data is logged autonomously via an on-board SD memory card (for later recovery). Alternatively, it can be interfaced with the
275 meteorological sonde via the XDATA protocol to transmit data in real time. During a Saharan dust event in Germany, UCASS was successfully dropped along with a meteorological sonde from the Dornier Do 128-6 research aircraft (Sec. 3.2).

**2.6 The radioactivity measurement sensor**

In case of nuclear incidents, vertical profiles of radioactive radiation together with meteorological- und geodata are of high relevance for decision makers and political authorities. Accurate detection of the boundaries of contaminated air masses are of
280 great importance for implementing airspace closures and taking appropriate measures on the ground. In such a case, the DWD is responsible for determining the hazard, both to carry out dispersion calculations and to assess the risk to air traffic. Up to now, airspace monitoring has been carried out using a Learjet 35A equipped with local dose rate measuring devices. An aircraft is well suited to be on site quickly in the event of a nuclear incident and can survey the relevant altitude range up to 14 km. A decisive disadvantage, however, is that considerable contamination of the aircraft and its occupants cannot be ruled out in the
285 event of an incident. The KITsonde system offers a suitable method, as the aircraft can fly significantly higher than the contaminated air mass over the dispersion area, and with the help of a dropsonde a high-resolution vertical sounding can be made. A series of sondes then provides a cross section through the affected air mass.

To perform these measurements, a miniaturised gamma dose rate measuring probe was developed by Scienta Envinet (2024) on behalf of the DWD, which uses the KITsonde container for release. A plastic scintillation detector in combination with a
290 silicon photomultiplier enables the recording of the gamma dose equivalent rate $H*(10)$ within 20% between 0.03-4 MeV with a temporal resolution of 1 s and allows statements to be made about the released radioactivity. To enable a near real-time evaluation at the DWD, the data has been retrieved and encoded in the aircraft and further submitted with a satellite modem (see Sec. 2.4) to the DWD data centre. The radioactivity sensor together with the meteorological sonde fits into the release container (Fig. 7a). After launching and delayed opening of the parachute the meteorological sonde separates from the release
295 container (Fig. 7b). The release concept is equal to the previously described payloads.



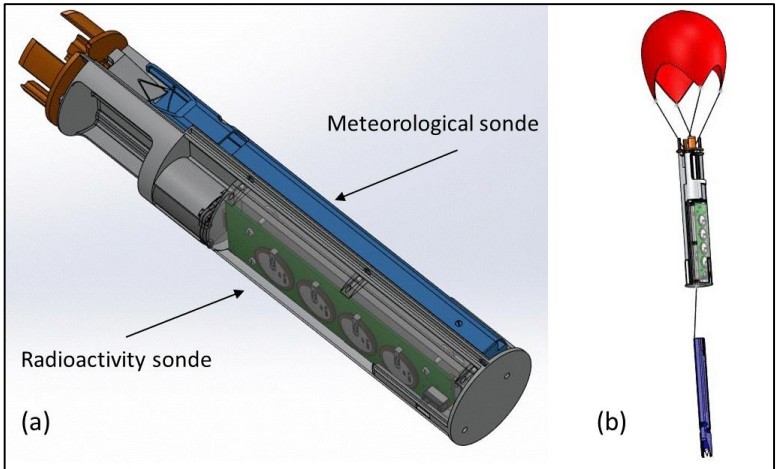

**Figure 7: (a) Release container with radioactivity sonde (green) and meteorological sonde (blue) and, (b) final sonde configuration during measurement flight.**

A test of the measurement system took place in October 2023 near Magdeburg, Germany. The probe was transported to the deployment area in a Learjet 35A and dropped from about 3 km height into a restricted airspace. The aim was to test the handling in the aircraft with initialising the sonde, the check of functionality of the dropsonde as well as of the entire data path via the satellite modem to the receiving station in the DWD data centre. The test was successful, although the transmitted data expectedly did not show any radioactivity besides cosmic rays.

**3. Observation results**

During the different stages of the KITsonde´s system development, test procedures were performed to evaluate the different system components (sonde configuration, data acquisition, receiver, release container, and meteorological sondes) and their proper interaction. This was done in the laboratory as well as during drop tests from different research aircraft, including launches from a turboprop Dornier Do 128-6 aircraft in the height range up to 6.5 km, from a Learjet 35A, and particularly

from HALO at 12 km height, for which the KITsonde was designed as part of the scientific payload. Comparisons covered different dropsondes of the type EL-18, different configurations of sondes within the release container, EL-18 dropsondes dropped from aircraft, and DFM-17 radiosondes operated simultaneously from the ground. From these comparisons, we found that the meteorological data are of the same quality as those of the frequently used Graw DFM-17 (Dirksen et al., 2024), the precursor model of the Graw EL-18 designed for KITsonde.

**3.1 Targeted observations with the Gulfstream 550 (HALO) jet aircraft during SouthTRAC**

HALO is the largest German research aircraft for atmospheric research (Krautstrunk et al., 2012). It is operated in major campaigns, each with a regional focus bringing together international teams with campaign-specific common objectives and a dedicated assembly of instruments. The "TRAnsport and Composition of the southern hemisphere UTLS campaign,





(SouthTRAC)" was conducted in the southern hemisphere in autumn 2019 to investigate the upper troposphere lower
stratosphere (UTLS) chemical composition and dynamics of the Antarctic vortex and gravity waves over the southern Andes
(Rapp et al., 2021).

The transfer flight section from Buenos Aires to Rio Grande (Argentina) on 06 November 2019 was used for performance
tests of the KITsonde system, in particular to demonstrate its capability of providing highly resolved spatial information on
humidity, temperature, and wind in a region with deep moist convection.

Synoptically, the day was characterised by weak pressure gradients and westerly winds of about 5-10 m s$^{-1}$ in the lower
troposphere (information available via search option from https://www.wetter3.de and based on GFS analysis data) increasing
to 20 and 40 m s$^{-1}$ in the middle and upper troposphere (Figs. 10 c,g), respectively. The wind at higher levels was associated
with the strong and close jet stream over Patagonia. A region of high convective instability developed over central Argentina
east of the Andes, as reflected by the lifted index (LI) and convective available convective energy (CAPE). At noontime, LI

was -4 K and CAPE reached values of about 1200 J kg$^{-1}$ (e.g., https://www.wetter3.de). Targeted towards the region of large
instability, i.e. where the development of deep convection was forecasted, an afternoon flight started in Buenos Aires and went
towards Cordoba (Fig. 8a). As expected, deep moist convection was present in the target region between 16:00 and 18:00 UTC
with radar reflectivity values of up to 44 dBZ in several cells (Fig. 8b). The region with moist convection moved slowly
eastwards in the course of the afternoon.

In total, 16 release containers with 60 meteorological sondes were dropped from about 12 km above mean sea level (AMSL)
in two sequences along transects from ESE to WNW roughly centred over the Sierras de Córdoba and crossing a line of
convective cells with strong radar reflectivity (Fig. 8a). The release containers were dropped at time intervals of approximately
1 to 2 minutes (corresponds to distances of 6 to 12 km, Fig. 8a). Due to the different parachute sizes, the mean fall speed of
the individual sondes varied between 3 and about 7 m s$^{-1}$ and the distances between the landing positions varied between only

a few and up to 30 km (Figs. 8a and 9).



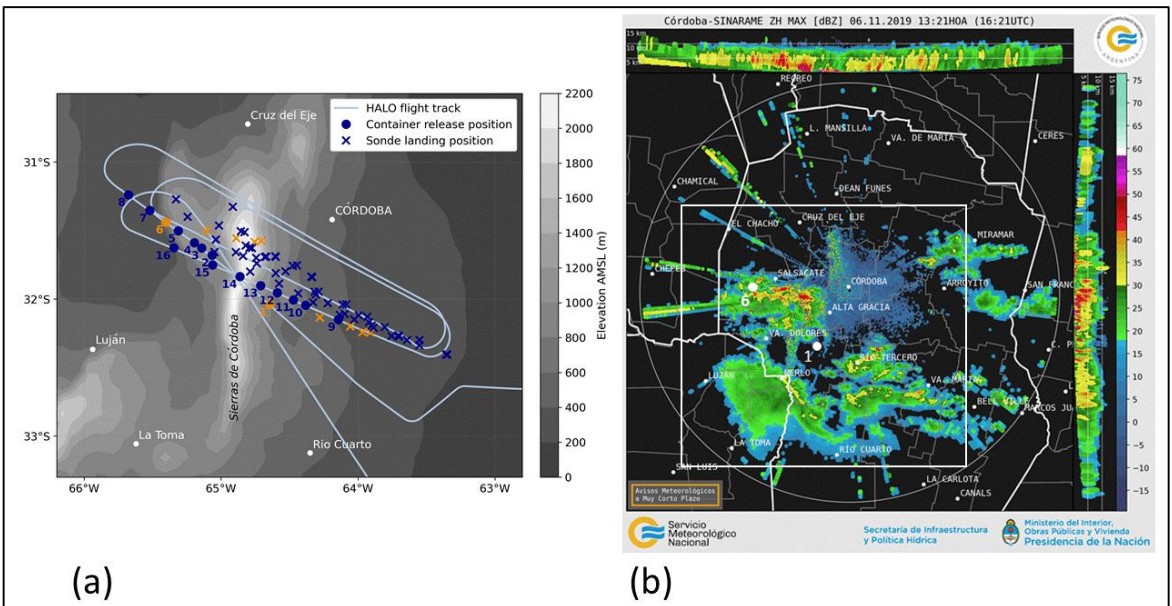

**Figure 8: (a) HALO flight track and release positions of the 16 containers (dots) that were dropped from HALO between 16:19 and 16:31 UTC (containers 1-8) and between 16:38 and 16:50 UTC (containers 9-16) on 06 November 2019. The positions of the releases of the containers and landing of the individual meteorological sondes are marked by blue and orange dots and crosses. The orange colored sondes are discussed in detail in the text. Grey shading represents the surface elevation above mean sea level (GMTED2010 elevation data; Danielson and Gesch, 2011); (b) radar reflectivity in dBZ at 16:21 UTC on 06 November 2019 of the radar at Cordoba operated by Servicio Meteorologico National of Argentina. Shown are projected views of maximum reflectivity from top (centre), from W (right panel), and from S (top panel). The white circles indicate the releases of containers 1 and 6 at 16:19 and 16:28 UTC, respectively. The white rectangle marks the area shown in (a).**

In order to visualise the spatial variability of the meteorological conditions on the different scales, we show the potential temperature anomaly and relative humidity of the meteorological sondes released from containers 1 to 8. For visualisation in the x-z-plane, the tracks of the falling sondes are projected onto latitude 32 °S.



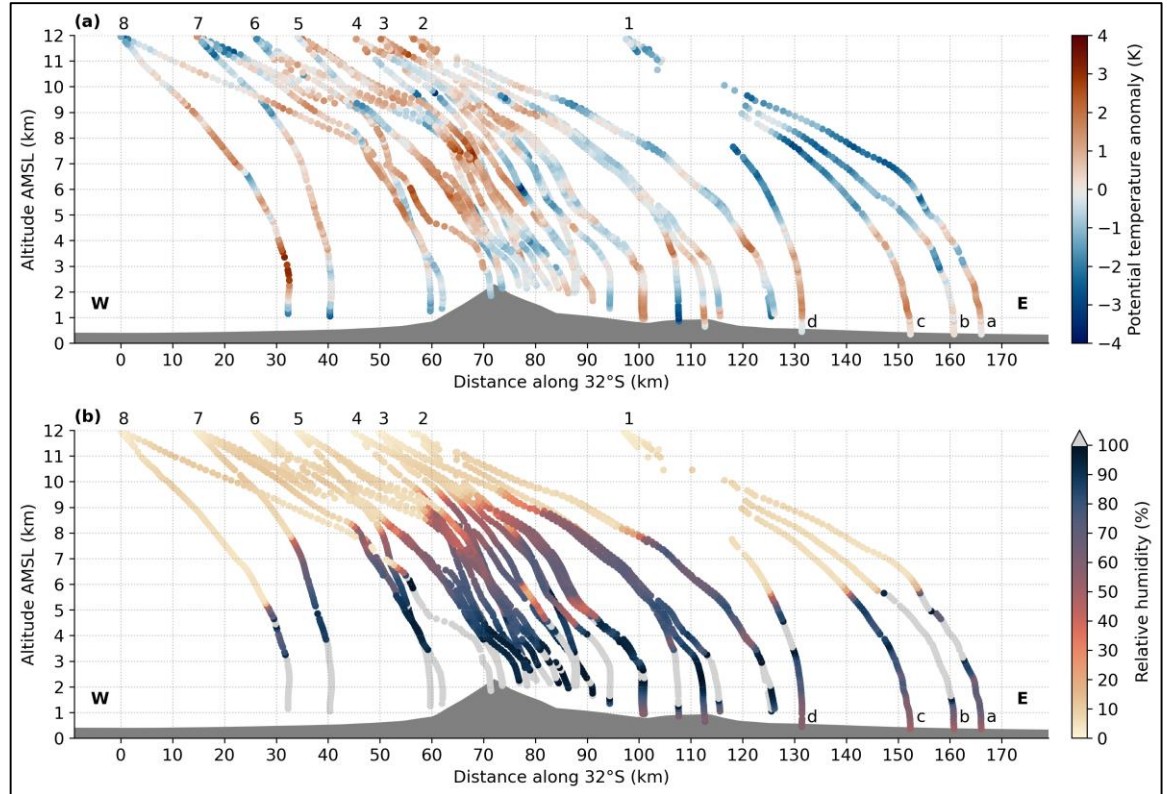

**Figure 9: (a) Profiles of potential temperature anomaly (differences from the mean of all shown profiles) and (b) profiles of relative humidity on 06 November of sondes dropped between 16:19 and 16:31 UTC. The numbers at the top of the panel indicate the release containers (corresponding to Fig. 8a). The cloud layers (relative humidity ≥ 100 %RH) are shaded in light grey. The distance indicated on the horizontal axis is the projection of the positions of the meteorological sondes onto latitude 32 °S. Dark grey shading represents the surface elevation along the transect.**

The spatial distribution of the temperature anomaly reveals several mesoscale structures. West of the Sierras de Córdoba, the temperature between 5 to 8 km AMSL is approximately 3-4 K higher than east of the mountain range. In lower layers, temperature anomaly in east-west direction tends to show an opposite sign with a local minimum above the Sierras de Córdoba (Fig. 9a). Relative humidity values east and far west of the Sierras de Córdoba are subject to a considerable decrease at heights above 5-6 km AMSL, while higher relative humidity values (e.g. 50 % RH) can be found over the mountain range up to an elevation of approximately 9 km AMSL (Fig. 9b).

Apart from these temperature and humidity structures over the whole zonal distance of approximately 150 km, individual sondes also reveal considerable differences. To demonstrate the benefit of using four meteorological sondes in one release container to observe mesoscale structures, we selected data from release containers 1 and 6 dropped from HALO on the first transect (Fig. 8a). Clear differences are observable in the profiles of potential temperature, relative humidity, wind speed, and



wind direction measured by the four sondes of container 1 (Figs. 10 a-d), which was released at 16:19 UTC. Due to the different

mean fall speeds and falling times (e.g. sonde 1b: 3 m s$^{-1}$ and 60 min; sonde 1d: 6 m s$^{-1}$ and 30 min), the sondes were distributed

over a total distance of about 37 km on the surface (Fig. 9). According to radar reflectivity data from 16:21 UTC (Fig. 8b), the

container was dropped into a region without precipitation, but the sondes drifted with the west-northwesterly wind (Fig. 10d)

into a region of convective precipitation (Fig. 8b).

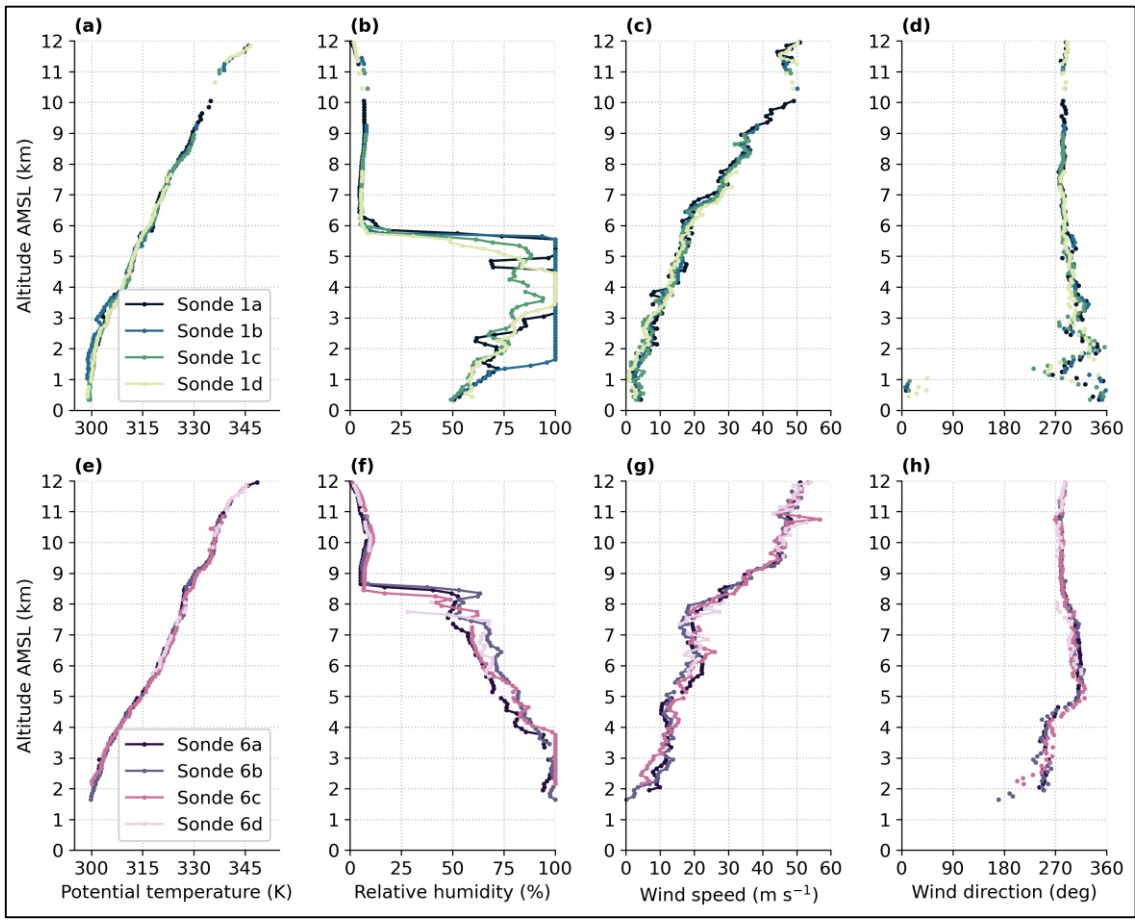

**Figure 10: Profiles of (a,e) potential temperature, (b,f) relative humidity, (c,g) wind speed, and (d,h) wind direction of the sondes dropped from the release containers 1 and 6 (see Fig. 8a) on 06 November 2019 at 16:19 and 16:28 UTC, respectively. All profiles are vertically averaged over 100-m intervals.**

The four temperature profiles in the area east of the Sierras de Córdoba show a similar vertical stratification (Fig. 10a). A well-mixed boundary layer reaches up to approximately 2 km AMSL and stable stratification is found above. Two more stable

shallow layers are present at about 6 and 8 km AMSL. The largest differences between the profiles of the sondes are obtained



for relative humidity (Fig. 10b). While sonde 1d measures the cloud top at about 5.5 km AMSL already, together with a weak temperature inversion (Fig. 10c), sonde 1b did not fall into a cloud at all. The cloud bases also vary considerably, i.e. between about 1.5 km (sonde 1b) and about 3 km AMSL (sondes 1a and 1d). Thus, the high thermodynamic variability in areas with deep convection is captured well by this kind of quadruple information – which cannot be provided by individual dropsondes

or single radiosondes. The horizontal wind speed varies between 5 m s$^{-1}$ at 2 km AMSL and 40 m s$^{-1}$ at 10 km AMSL (Fig. 10c) in all profiles. The wind in the boundary layer shows some veering from northerly to westerly winds. It is predominantly northwesterly between 2 and 6 km AMSL and westerly above (Fig. 10d). This vertical profile of the horizontal wind is visible in all four soundings.

The sondes of container 6 were released at 16:28 UTC west of the Sierras de Córdoba and separated by about 20 km when

they reached the ground (Fig. 8a). As they were transported southeastward with the prevailing northwesterly wind (Fig. 10h), they reached the area of the Sierras de Córdoba. This is also indicated by their landing positions between 1.5 and 2 km AMSL elevation (Figs. 10e-h). Despite some smaller differences in e.g. wind direction close to the surface (Fig. 10h), thermodynamic and dynamic conditions are quite similar even for relative humidity (Fig. 10f). However, considerable differences of all meteorological parameters are found compared to the conditions east of the Sierras de Córdoba (Figs. 10a-d). The boundary

layer over the Sierras de Córdoba is more stable and the mid-troposphere is warmer than east of the mountain range. The relative humidity indicates that the clouds do not extend higher than 4 km AMSL. However, relative humidity is enhanced up to about 8-9 km AMSL (Fig. 10f), i.e. much higher in the upper troposphere than east of the mountains (Fig. 10b). The comparison of the wind fields below 4 km AMSL also reveals some clear differences. While the wind over the mountains between 2 and 4 km AMSL is from the west (Fig. 10h), it is from northwest east of the mountains (Fig. 10d). The wind at

about 1 km above ground level (AGL) predominantly is from northerly direction east of the mountains. Over the Sierras de Córdoba, by contrast, two sondes indicate southerly winds and two sondes show more westerly winds.

We conclude that four meteorological sondes released from the different containers captured the spatial heterogeneity of dynamic and thermodynamic conditions on the meso-gamma scale. This is an added value of the KITsonde system especially when high-resolution information is important. Confirming the technical maturity and usability of the KITsonde system, we

found that all 60 meteorological sondes worked reliably and provided usable data from their release points down to the Earth's surface. The operational procedures for observations with the dropsondes launched from HALO were sufficiently flexible to allow targeting an area with a large mesoscale convective system.

**3.2 Observations with UCASS from Dornier Do 128-6 during a Saharan dust layer event**

The KITsonde system, coupled with an optical particle counter (OPC) UCASS, was tested on 01 and 03 August 2013 near

Magdeburg, Germany, within the ED-R74 restricted airspace centred approximately on 11.45 E longitude and 52.45 N latitude, called the Colbitz-Letzlinger Heide. Fortuitously, a tropospheric Saharan dust layer was present during the second day. The Dornier Do 128-6 research aircraft was equipped with the KITsonde cabin installation. Release containers held single meteorological sondes together with an UCASS counter. Drops took place typically from an altitude of 6.5 km AMSL, with





up to four drops per each one flight on two flights. After release, the meteorological sonde and the OPC separated, but remained
tied together by a 1 m long polyester line, with the meteorological sonde located lowest, followed by the OPC, and the
parachute attached at the top of the "train" (Fig. 6d). This configuration creates a double pendulum which tends to dissipate
kinetic energy and helps to reduce swinging motions of the OPC, thus aligning it with the air velocity vector for correct particle
sampling (Smith et al., 2018). OPC data was stored on an integral memory card. The sondes were recovered after the flights
based on the meteorological sonde's GPS position information. To ensure precise position data even near ground level, the
aircraft remained airborne and in the vicinity for good telemetry reception. Following recovery, most of the sondes were
deployed for the second time during the second day.

For brevity, we focus mainly on results from two drops which took place on 03 August 2013 at 13:10 and 13:13 UTC, with
OPCs "U3" and "U4", respectively. Saharan dust was present aloft, as verified by independent modelling and AERONET sun
photometry retrievals nearby, being shown below in Fig. 13 and 14. For comparison, we also show a profile from OPC U3 on
01 August, when dust was absent. The sonde "trains" descended at mean velocities of 5.7 and 5.1 m s$^{-1}$ for sonde U3 and U4,
respectively, the difference being probably due to slightly differing parachutes.

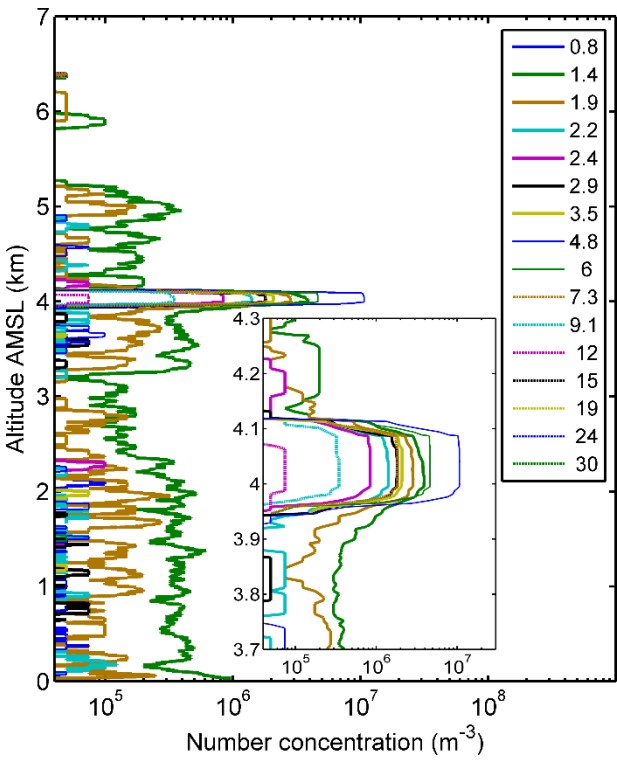

**Figure 11: Vertical profile of aerosol concentration during the Saharan dust episode as measured by sonde U4 in 16**
**size bins, with centre diameter (□m) shown in the legend (03 August, 13:13 to 13:33 UTC). individual concentrations**
**are averaged with a rectangular window width of 20 (i.e. 20 s, equivalent to about 100 m). An embedded shallow**



**cloud layer is visible at 4 km altitude AMSL - inset. See text for further details. Data reprocessed, based on Smith et al. ( 2019).**

The UCASS provides very rich and fine-grained raw data. It shows a vertical profile of particle number concentrations measured in 16 individual size bins during the Saharan dust episode on 03 August (Fig. 11). For easier interpretation, quantities
integrated over particle size can be displayed, such as effective radius (or diameter), liquid water content (LWC) for clouds, or volume content for dust. Fig. 12 shows the effective diameter together with the aerosol volume content measured during the Saharan dust episode. In addition to coarse aerosol in the boundary layer and the free troposphere, the plots reveal an embedded cloud layer just above 4 km altitude AMSL, with the highest number concentration found in the 4.8 ☐m bin at 20 ml$^{-1}$ and a total maximum concentration of 144 ml$^{-1}$. The meteorological sonde indicates a relative humidity peak centred on the cloud
layer, with a maximum value of 95 %RH and a temperature of -1°C. The effective radius in the cloud layer is 5.7 ☐m (Fig. 12). These two findings together suggest that the cloud, was dissipating. Comparison of the data for 03 August with a similar profile obtained on 01 August before the dust episode shows that the Saharan dust was most prominent in the free troposphere, as much lower aerosol concentrations were found on the earlier date. The coarse aerosol at lower levels below about 2 km AMSL likely originated locally from the sandy, dry heath (Colbitz-Letzlinger Heide) over which the dropsonde tests were
carried out (Fig. 12).

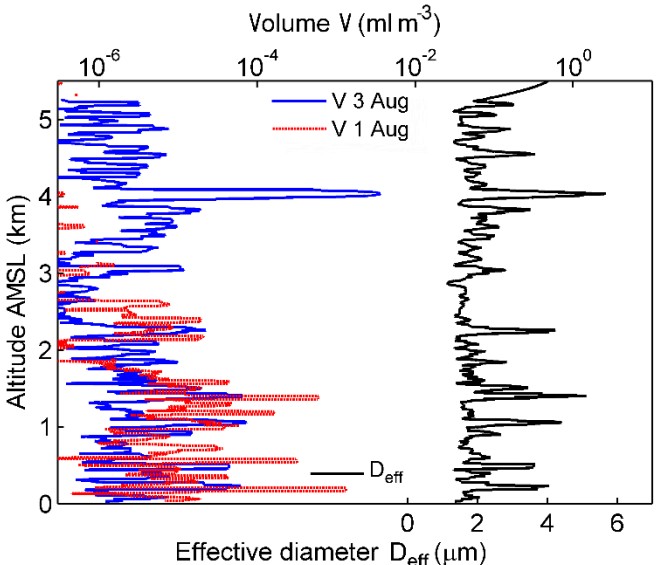

**Figure 12: Effective diameter and aerosol volume concentration measured during the Saharan dust episode on 03 August (sonde U4) in comparison with the volume concentration on 01 August (no Saharan dust, sonde U3).**

Alternatively, size distributions integrated over (narrow or wide) sections of altitude provide useful information. An example
is given in Fig. 13, which shows size distributions in integrated, 400 m thick layers during the Saharan dust episode on 03



August. The distribution in the 4.0 to 4.4 km layer clearly stands out. It is associated with the cloud layer seen in the high-resolution vertical profile in Fig. 11.

Another example is given in Fig. 14, which compares the size distribution measured by the sonde U4 on 03 August during the dust episode with an almucantar remote sensing retrieval from the AERONET (AErosol RObotic NETwork, Holben et al.,
1998) sun photometer in Leipzig, 100 km SSW of the drop location. This comparison is an example of how such in-situ atmospheric profiling can be used for validating remote sensing techniques, while accounting for atmospheric composition and avoiding contamination from clouds, which hinder aerosol retrievals.

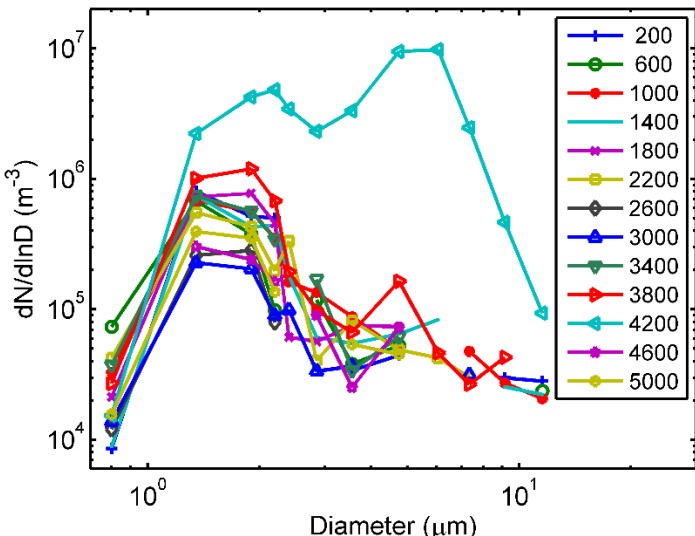

**Figure 13: Particle size distributions integrated over 400 m thick layers measured by the sonde U4 during the Saharan dust episode**
**on 03 August. Layer centre altitudes (m) are shown in the legend. The anomalous distribution is due to a thin cloud layer just above 4 km AMSL altitude.**



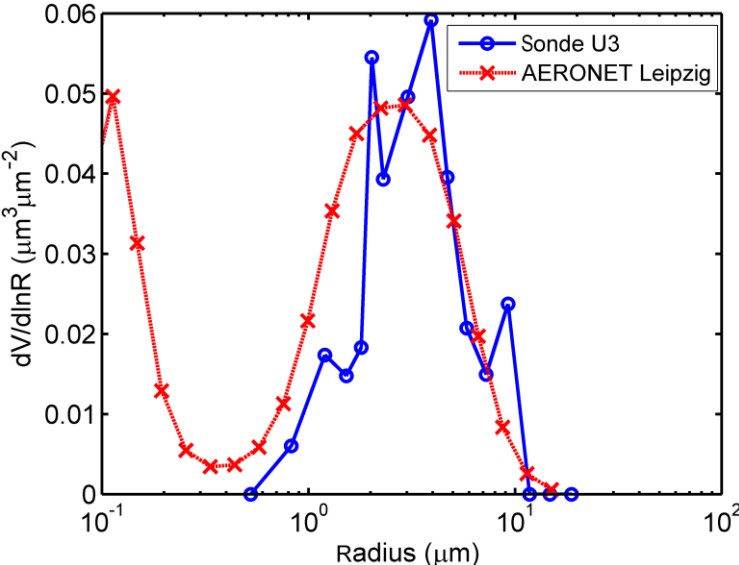

**Figure 14: Aerosol size (radius) distribution measured between 3 and 5 km AMSL altitude by the sonde U3 on 03 August at 13:17 UTC during a Saharan dust episode in comparison with the Leipzig AERONET sun photometer retrieval level 2.0 at 15:34 UTC.**

### 4. Conclusions and outlook

Tthe KITsonde system meets aöö expectations in terms of quality, completeness, and usability. The versatile release container of the KITsonde system is designed for use with the standard RD93 dispenser of widely used research aircraft's AVAPS. Up to 4 meteorological sondes can be dropped with one release container and 30 sondes (corresponding to 8 release containers dropped) can be received at the same time. The different sizes of the parachutes allow for controlling the mean fall speed of the meteorological sondes between typically 3 and 7 m s$^{-1}$ and the spatial separation of the sondes after release from the release container, in particular by using model–based pre-launch wind information. On the Earth's surface, the sondes from one release container may be separated by up to about 50 km in strong-shear weather situations, as encountered in our case study. Using paragliders instead of parachutes may even allow more flexibility in sonde trajectories such as circling. For a further increase in the number of meteorological sondes in the release container of the KITsonde system, the use of swarm sondes without parachutes (e.g. SkyFora, 2024) could be a reasonable option in future.

The advantage of the novel KITsonde system is its capability of dense spatio-temporal probing of fast developing atmospheric phenomena, such as heterogeneous boundary-layer structures, thunderstorms, mesoscale convective systems, or frontal processes. It can be used in a single sonde configuration in areas before and behind such phenomena, where horizontal homogeneous distributions of atmospheric conditions can be expected, or in a multi-sonde configuration within and close to atmospheric phenomena of high variability in space and time.

In addition to a purely meteorological application, the release container payload can be configured as a cloud and/or aerosol "particle spectrometer" or as a radioactivity sensor. In these cases, the measurement system also includes a meteorological sonde. Both extensions open the applicability of the release container concept for air surveillance tasks. Vulcanic eruption as



well as nuclear incidences with radioactive exposition require a system that can be allocated quickly in the region of interest,
e.g. in the European airspace. Highly resolved spatial measurements are required in such cases without penetrating the airspace
with in-situ sensor techniques aboard the aircraft itself. The knowledge of the spatial extent of volcanic ash or radioactive
contamination is important to minimise the impact of airspace closures by air traffic control. The transfer of delicate data to
national authorities can be achieved by satellite communication on a near real-time basis.

The applicability of the meteorological sonde is demonstrated for a weather situation with several thunderstorms over
Argentina. As expected, measurements reveal considerable variability in humidity, temperature and the horizontal flow. Future
deployments in convective situations would benefit from good spatio-temporal attribution of sonde locations relative to the
convective subregions, which may be based on precipitation radar data and satellite information.

The inclusion of the optical particle counter UCASS provided simultaneous measurement of the vertical size distribution
profile of atmospheric mineral dust or cloud droplets. Tests presented here demonstrate that the soundings do not only yield
the profiles of coarse aerosol, but also allow for the identification and characterisation of embedded cloud layers. This
conclusion is supported by cloud profiling tests carried out using a physically similar version of UCASS mounted on a remotely
piloted multicopter (Girdwood et al., 2020). The KITsonde coupled with UCASS measurements can be useful for validating
remote sensing techniques, aerosol dispersion or cloud models, as well as for studying cloud processes in general. The vertical
profiling capability, in contrast to the essentially horizontal one when aircraft-mounted probes are used, is an advantage in the
context of both remote sensing and cloud processes. Remote sensing typically involves vertical profiling (or integration) and
cloud related processes are determined predominantly by processes like convection, sedimentation, and precipitation acting in
the vertical direction.

As the meteorological sonde with the radioactivity sensor has been tested successfully, it is now planned to use them in the
MEASURE project (development and testing of a drone-based system for in-situ measurement of volcanic ash and
radioactivity to maintain aviation safety; MEASURE, 2024) funded by German Federal Ministry for Economic Affairs and
Climate Action (BMWK). One aim of MEASURE is to assess the benefit of vertical dropsonde profiles in aviation consulting
processes. This includes the development of strategies and products as part of the operational processes of DWD. Additionally,
the data could be used for application-oriented supply of dispersion calculations and, if necessary, a concept for validation
based on measurement data.

Future deployments of the KITsonde system on HALO are planned during the Arctic Springtime Chemistry Climate
Investigations (ASCCI) campaign in March 2025 to test the transmission of real-time data to the Global Telecommunication
System (GTS) for operational data assimilation. This deployment during ASCCI is a preparatory step towards the North
Atlantic Waveguide, Dry Intrusion, and Downstream Impact Campaign (NAWDIC; https://www.nawdic.kit.edu/) in January
and February 2026, where targeted observations of coherent air streams and mesoscale dynamics of extra-tropical cyclones
over the North Atlantic will be performed with the KITsonde system.



**Competing interests:** The contact author has declared that none of the authors has any competing interests.

**Data availability:** Data from observational campaigns are provided on request.


**Access to KITsondes**: Potential users of the KITsonde in present or new configurations shall address enviscope GmbH.

**Acknowledgements:** We thank the crew of the Dornier Do 128-6 research aircraft D-IBUF of the Technische Universität Braunschweig Helmut Schulz and Mark Bitter as well as the HALO operational team of DLR for their very helpful support.

Funding by DFG SPP 1294 "Atmospheric and Earth system research with the High Altitude and Long Range Research Aircraft" (HALO) under project numbers 47468573 and 179877336 and by the Helmholtz Association is acknowledged. We also acknowledge the team at University of Hertfordshire, led by Paul Kaye, which developed and built the UCASS sensors. We are indebted to the AERONET network for providing aerosol retrievals and particularly to Albert Ansmann for operating the sun photometer at IfT Leipzig. Finally, we are grateful to Maike Schröder from KIT for English correction of the manuscript.

**Author contributions**: CK, UC, AW and NK developed the idea of the KITsonde. FS, JF, FR, and AW designed the meteorological sonde. DS and HF integrated the KITsonde components, designed the release container with adapted payloads as well as performing the certification of the system according to airworthiness standards. DS programmed the near real-time data retrieval and the graphical interface for the in-flight visualisation of the data. ZU led the development of the UCASS sensor and its deployment for the KITsonde, and processed and interpreted the optical particle measurements. DE was 535 responsible for development and integration of the radioactivity measurement system. TF, RH and AW were responsible for the implementation of the KITsonde system for observations and conduction of measurements with the Dornier Do 128-6. PG was responsible for the aircraft measurements with the KITsonde system over Argentina. CK, NK, UC, and BK analysed and described the measurements with the meteorological sonde performed over Argentina. CK and NK prepared the article with contributions by all co-authors.

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
