# Peer review of "A New Versatile Dropsonde for Atmospheric Soundings – The KITsonde"

_EGUsphere, 2024_

## Author Comment (AC3)

The authors thank the reviewer for the well thought out and constructive comments on the manuscript. All replies to these comments are inserted below (in yellow) and information is provided, how the manuscript has been changed accordingly (in green). In the text the new parts are highlighted in grey.

**General Comments**

The paper describes a new dropsonde system – the KITsonde, which has novelties such as (1) multiple sounding profiles from a single launch, (2) a modular payload design thanks to a release-container concept and (3) reception from up to 30 channels simultaneously. The paper is well-written and coherent. The structure makes sense, starting with a background on dropsonde development, detailed description of the system itself, the different possible configurations and the tests done in the field. Overall, the measurement techniques described herein will improve the type of atmospheric soundings being taken currently as well as the aircraft strategies being employed. The KITsonde holds exciting potential for advancing atmospheric observations especially with the three aforementioned novelties.

However, I do have a few comments, which I wish the authors would clarify.

1. The current design is compatible with dispensers of RD-series dropsondes, which is great for integrating with existing aircraft systems. However, these dispensers are being phased out as NCAR and Vaisala transition to the NRD series (e.g., NRD-41). The newer, smaller sondes require research aircraft teams (incl. HALO) to modify or replace dispensers, often adopting the automated launcher systems. This is a crucial point because a smaller release container for the KITsonde would impact the first two novelties I list in the general comments. As the older dispensing systems are becoming obsolete, the KITsonde's compatibility with them is no longer an advantage. Could the authors address the KITsonde's compatibility with the new dispensers, either for HALO specifically or more broadly?

We agree with the reviewer's remarks. The consequences are, however, only, that the KITsonde cannot be used with aircraft solely offering the smaller dispenser. Present aircraft have the necessary opening for the larger dispenser and – as confirmed for HALO - a kind of adapter will be used to reduce the open diameter (the hole) within the aircraft skin. Both the old and new dispenser can be alternatively used even in the future. New aircraft may offer dispensers only for the new form factor. For those cases, single EL18 dropsondes are small enough to be dropped.

The raised issue is addressed in the future outlook section as follows (lines 537-540):

With a gradual transition to smaller dispensers for the NRD series (e.g., NRD-41) by NCAR and Vaisala, only single EL18 dropsondes are small enough to be used. Since present dropsonde-releasing aircraft have the necessary opening for the larger dispenser, a kind of adapter, as confirmed for HALO, will be used to reduce the open diameter (the hole) within the aircraft skin. In this case, both the old and new dispenser can be used in the future.

2. The variability in the parachute design is unclear to me and the paper will benefit from better description. For instance, phrases like "individually sized parachutes" (L97) and "different sizes of the parachutes" (L469) tell me that parachutes may vary in size to maintain separation in vertical space. If so, what are the possible

sizes and configuration options? This information is critical for planning the launches because sounding strategies depend on the descent rate, which affects how closely the sounding trajectory approximates a vertical profile as well as how closely the sounding approximates to an "instantaneous" profile. For the KITsonde, in a 4-sonde release container, what parachute sizes are used and what are the expected descent rates for the 4 sondes, and is this configuration standard or could the user choose? Similarly, for a single-sonde release container (with or without other payloads), what is expected descent rate? H

We agree with the comment. The description of the parachutes is now extended (including some wording of the reviewer) and an analysis of the fall speeds for the different sizes has been performed, and a related table of results is added to the manuscript as a new subsection 2.3.2:

**2.3.2 Parachutes**

The parachutes are chosen to determine the descent rates, which affects how closely the sounding trajectory approximates a vertical profile as well as how closely the sounding approximates to an "instantaneous" profile. The parachutes of 64 cm in diameter being used are manufactured by BBL Elektronik & Aeromet GmbH (https://www.meteorologyshop.eu/en/balloons/radiosonde-balloons/368/meteorological-parachute-pc-055). To allow for different fall speeds, the effective areas of the parachutes were reduced by cutting away outer trapezoidal sectors of fabric between the holding lines. Tab. 2 shows the effective diameters, corresponding to the effective area of the parachute. Fall speeds versus parachute size were analysed for the SouthTRAC campaign (see Sect. 3), where 60 meteorological sondes EL-18 without any additional sensor or communication electronics were used.

Table 2: Mean fall speed and standard deviations in m s$^{-1}$ of meteorological sondes dropped with parachutes of different effective diameters. The fall speeds are given for different height ranges during the SouthTRAC campaign.

| Height range (km) | Parachute effective diameter (cm) in line 2 and area (cm$^2$) in line 3 for the meteorological sonde EL-18 of 73 g weight | | | |
|---|---|---|---|---|
| | 64 | 40 | 32 | 24 |
| | 3217 | 1257 | 804 | 452 |
| 8 - 12 | 4,5±1,2 | 7,0±2,1 | 8,1±2,1 | 11,2±1,6 |
| 4 - 8 | 3,3±1,0 | 4,9±1,5 | 5,9±1,4 | 8,7±1,0 |
| 0 - 4 | 2,6±0,6 | 3,8±1,2 | 4,8±1,1 | 7,3±0,5 |

The parachutes for the release container, for the coupled radioactivity/meteorological measurements, and for the particle/meteorological measurements are more robust and of a x-pentamine shape. They are manufactured by Spekon Co.

([https://spekon.de/seilschirme.html](https://spekon.de/seilschirme.html)) and consist of 5 quadratic sections, each 15 x 15 cm. The total area is 1125 cm$^2$.

3. This third point relates slightly to the previous one in terms of descent rate. I am not completely convinced by the motivation for the satellite modem configuration. Currently (e.g. Ehrlich et al 2024, https://doi.org/10.5194/essd-2024-281) HALO dropsondes are sent to GTS in near real-time, i.e. as soon as the dropsonde makes a landing and therefore can also be sent to ground-support. Generally, the difference in real-time and near real-time is around 12-15 minutes and should not affect data assimilation too much. Of course this changes if the descent time is close to 45-60 minutes, which means that the aircraft telemetry could be out of range. But this advantage of the satellite telemetry then comes at the cost of a multi-sounding launch, which provides the novel spatio-temporal density I would argue is the best feature of the KITsonde. It does work as an example of different payload capabilities, but I struggle to find a practical use-case for it where it is advantageous over the conventional sondes, such as what the UCASS and radioactive payloads demonstrate. I would appreciate the authors' clarification here.

The arguments are generally correct, as far as that the option of satellite communication is not highly necessary or a big advantage to available systems, because it comes at the cost of a multi-sounding launch. The satellite link was tested only once and is not in central focus. It was developed to allow for small descend speeds, when a fast aircraft like HALO may lose the communication link. It also maintains future options of dropping sondes from normal, non-research aircraft without a signal receiving unit. NRT assimilation of very high spatio-temporal multiple dropsonde data like KITsonde are anyhow questionable for operational DA. A „one sonde per container" is preferable for that case, and KITsonde would provide comparable information to the NCAR/VAISALA system.

We modify the sentence in line 297-301:
"Using the satcom module is only recommended for cases when the descent of the sonde is slow, causing a descent time of 45-60 minutes, and the aircraft would need circling to keep telemetry contact. The satellite communication allows only one meteorological sonde to be dropped with the release container (Fig. 1) so that it comes at the cost of a multi-sounding launch. The satcom module would also allow for dropping sondes from normal, non-research aircraft without a signal receiving unit."

**Minor comments**

Title: Why HALO specifically when the system has already been demonstrated with 2 other aircraft too?

The use on HALO is the final goal, and the development was performed and mostly funded in the context of the HALO consortium.

But we agree, that the KITsonde is currently used from other aircraft and may be even used more in the future, and drop HALO from the title.

L45-47: Acronyms are not defined

Acronyms are checked and modified as follows (new lines 45-49):

Early attempts to use dropsondes with the capability to measure wind based on the very low frequency radio navigation system OMEGA were made in the 1970s, continued by also using the navigation system LORAN-C in the 1980s. In 1994, the American National Center for Atmospheric Research (NCAR) and National Oceanic and Atmospheric Administration (NOAA) as well as the German Aerospace Center (DLR) agreed to develop a GPS dropsonde (RD93) based on Vaisala radiosonde technology.

L49 : Unclear why the horizontal spacing of 100 km between drops? Is there a reference for explanation?

The cited spacing of 100 km is from grey literature, which – when checking again – could not be found for proper citation.

We remove the part of the sentence "with the telemetry system allowing for a horizontal spacing of 100 km between drops."

L52 : NWS here is particularly the US National Weather Service as opposed to different countries' weather services when defined in L44.

Checked and modified by replacing NWS by US National Weather Service

L81: What type of flexibility in operations?

Checked and modified by deleting "in operations" because it is clear that the flexibility refers to aircraft or platforms.

L99-100: This is an excellent and very pragmatic advantage in favour of having the release container concept. :)

L157 : Is there a reference for 250 m/s? I believe 200 m/s might be closer to cruising speed during flight operations, but I am not familiar with all HALO payload configurations.

We had a view into the SouthTRAC data. For flight sections ST23a und b the average „Speed over Ground" was 193 and 180 m/s,  and „True Air Speed" was 242 and 236 m/s.

We replace 250 m/s by 240 m/s.

L194: Please mention the weights and CoG for the standard 4-sonde configuration?

We added: The container with 4 EL-18 sondes has it's centre of gravity at 195 mm from the bottom. The weight of the release container with 4 sondes and parachutes is 761 g.

L207: How many minimum satellite connections are needed for this?

We assume the comment refers to "The cold-start time (time to first fix) is 26 s under good conditions (open sky)."

Time to first fix is primarily dependent on acquisition time and the time the receiver needs to obtain enough of the almanac and ephimeris to be able to provide a valid navigation solution. The latter is also dependent on the number of satellites received. The manufacturer does not specify which number of satellites needs to be in view for the 26 s figure in the datasheet to be valid. However, experience shows that a time of less than 30 s can usually achieved with 10-15 satellites in view. The navigation engine itself has a 72 channel receiver.

Slightly extended text (lines 220-225):
"For wind and position retrieval, the GNSS receiver u-blox MAX-M8C with 72 channels is used. It can receive various combinations of the L1 signals from GPS, GLONASS, and Beidou." …..  "Experience shows that a time of less than 30 s can be usually achieved with 10-15 satellites in view."

L210: PTU 1.12 s (and wind 1 s)… Discrepancy with the abstract, where 1.2 s is mentioned for both.

1.12 s and 1 s are correct and values are corrected in the abstract.

L230: Humidity above 100% RH stated, but table shows

We cannot claim to measure supersaturation quantitatively, so we correct the wording.

Replaced in Table 1 „Range of suitable measurements" | "0 – 100 %RH

L232-233: It is unclear why the heated sensor is not used and why it is less relevant than to radiosondes. Could riming not be a case for when heated sensors prove to be useful?

The dropsonde measures and reports the temperature of the humidity sensor. As the temperature sensor cannot be collocated on the humidity sensor, there is a small space between the two sensors, which leads to a larger uncertainty component than with the heated sensor. Nevertheless, the temperature data can be used to correct the humidity measurement for the temperature of the humidity sensor. We have rephrased the sentence to highlight this better. This setup is also able to detect situations, where the humidity sensor might experience more wet-bulbing than the ambient temperature sensor. The main difference between the two sensors for cloud-exits is the faster response time of the heated sensor. However, in the dropsonde application, response time decreases during flight progression, inverse to radiosonde operation.

Modified text (in lines 240-247):

The humidity sensor uses the same type of polymer but is unheated and has its accompanying temperature sensor located a few mm apart (Fig. 5b). The heated humidity sensor of the latest DFM-17 generation was not yet available at the time when the EL-18 was designed. The close temperature sensor can be used to correct the humidity measurement for the humidity sensor temperature. Still, this slightly reduces dynamic performance and response time, especially after the sensor has experienced precipitation. However, as the sensor response time decreases during the sounding, inverse to radiosonde operation, this problem is not as severe as with radiosoundings. As for most radiosonde algorithms (Dirksen et al. 2024), an active clipping of measured relative humidity values > 100 %RH is performed. Humidity values before clipping of up to 120% could be observed, which is a typical value.

L248: Are the interruptions often enough to justify an SD card buffer (thereby increasing waste and expense per sonde)?

During the tests the interruptions were frequent enough to justify a SD buffer card, which costs only 4 €.

We added in lines 296-297: This buffering system is necessary to avoid data loss when the connection to the satellite is interrupted. The data losses happened during the tests.

L310-312: Have these comparisons been documented somewhere? Could references be provided?

We wanted to mention that numerous tests and checks have been done in all stages of development. They were partly just for functionality and mostly only qualitatively. The results took influence on modifications and improvements. No systematic documentation has been done and no citable references are available.

L351: anomaly with respect to what?

As already written in the figure caption we add on that and clarify the sentence as (line 415): ".... we show the potential temperature anomaly (differences from the mean of all shown profiles) and ....."

L368: I believe "clear differences" is questionable phrasing except in the case of humidity.

Comment is accepted.

As we discuss the differences in the subsequence sentences in detail, the sentence is modified (in lines 434-435):
"The differences in the profiles of potential temperature, relative humidity, wind speed, and wind direction measured by the four sondes of container 1 (Figs. 10 a-d), which was released at 16:19 UTC, are as follows. "

L382: Shouldn't it be sonde 1c instead of 1b? For me, 1b looks like it went through the deepest cloud layer.

That is right and corrected as proposed.

L403: meso-gamma… Is it per Orlanski (1975)? Please define spatial extent or provide suitable reference.

Yes, we referred to Orlanski.

We add the scale and Orlanski in the text (lines 468-469): "…… captured the spatial heterogeneity of dynamic and thermodynamic conditions on the meso-gamma scale (2-20 km, Orlanski, 1975)."

Orlanski, I.: A rational subdivision of scales for atmospheric processes. BAMS, 56, 527-530, 1975.

L423: Where is the "independent modelling" part in Figs 13 and 14?

The "independent modelling" included the MACC, NAAPS and BSC-DREAM8b models, among others. Unfortunately, some of the model data appears to be ephemeral and in the intervening decade has ceased to become publicly available. Two exceptions are the Barcelona Dust Regional Center and NAAPS archival datasets – see e.g. the optical depth data for 3$^{rd}$ August 2013: https://www.nrlmry.navy.mil/aerosol/globaer/ops_01/europe/201308/2013080306_globaer_ops_europe.gif.

We have inserted the following explanatory text (485-487):
"including the MACC, BSC-DREAM8b and NAAPS models, the latter two archives available at the Barcelona Dust Regional Center (2025) and Naval Research Laboratory websites (2025), respectively."

and the references:
Barcelona Dust Regional Center, Products: https://dust.aemet.es/products/, last access: 20 January 2025.
Naval Research Laboratory, NAAPS (Navy Aerosol Analysis and Prediction System): https://www.nrlmry.navy.mil/aerosol_web/, last access: 20 January 2025.

L444: Is there a suitable reference for such coarse aerosol origins locally at 2 km altitude?

The authors are not aware of any such published descriptions, the remark in question is based on some visual observations made during the recovery of the dropsondes, such as during sunset, when such a layer seemed to partly shade the sun disk.

Figure 14: Why is there a 2-hour difference between the compared measurements? And why was the 3-5 km altitude window chosen?

AERONET size distribution retrievals can only be made when full almucantar measurements can be carried out, which can only happen when the sky is sufficiently clear of clouds. Hence there can be wide temporal gaps between retrievals. Fortunately, in this case a successful measurement at the Leipzig AERONET site was made only about two hours after the sounding. The altitude window was chosen for the comparison of sonde U3 with the AERONET retrieval to include just the purported Saharan air layer, as indicated by discontinuities in both number concentration (not shown for sonde U3 but also visible for sonde U4, Figs. 11 and 12) and humidity profiles, especially to the exclusion of the boundary layer dust that was assumed to be of local origin specific to the drop area, as stated in lines 440-444.

No related text modification. An error was spotted in the text, line 452: "sonde U4 on 03 August" should be "sonde U3 on 03 August". The caption to Fig. 14 that the text refers to correctly gives "sonde U3".

L466: Typos in the first few words

Corrected: The KITsonde system meets all expectations

At multiple places, figures are not numbered in the same sequence as their appearance in the text.

The text was checked accordingly and figures are now numbered in the sequence of their appearance.

---

## Author Response (AR1)

The authors thank the reviewer for the critical and competent comments on the manuscript. All replies to these comments are inserted below (in yellow) and information is provided, how the manuscript has been changed accordingly (in green). In the revised text the new parts are highlighted in grey.

**Summary:**

The manuscript by Kottmeier et al. describes the KITsonde dropsonde system, which can release 4 sondes simultaneously inside a dedicated launch container. This allows dropping either meteorological sondes of the same type, or sondes measuring different parameters such as cloud particles and gamma radiation. This system also allows direct transmission of data to the ground via a communication satellite.

This manuscript discusses the details of the dropsonde system and shows data from several field campaigns to demonstrate its capability.

The manuscript is overall well written. Some details require more discussions and clarifications, and I can recommend publication of this manuscript after some minor modifications following comments listed below.

**Detailed comments:**

Section 2.2: How long does the release of the individual sondes from the release container take and how much time or altitude below the aircraft is lost before measurements can be considered reliable as a result of this delay?

The delay time consists of three parts: (i) the programmed delay between the detection of a release from the aircraft and the begin of the container opening, (ii) the time for opening the container, and (iii) the time needed by the sensors to adapt to the surroundings, and, respectively for the wind, the time needed by the GPS-receiver to provide georeferenced position data. For wind, experience shows that a cold start time to first fix of less than 30 s can be usually achieved with 10-15 satellites in view. The total delay of (i) – (iii), in accordance with analyses of SouthTRAC data is less than 30 sec, corresponding to 500 – 1000 m at 12 km height. For wind, the total delay time may be up to 60 sec.

An example is given in the figure below. Obviously the meteorological sensors seem to adapt to the ambient air immediately after leaving the aircraft, reducing the height range of „lost" data below the aircraft to about 500 m. The equivalent height range for wind is closer to 1000 m.

[Figure]

In the revised version (lines 179-181) we added "In agreement between preset delays and profile analyses, the total time to get reliable temperature, humidity and pressure is less than 30 sec, respectively 500 m for HALO when dropping from 12 km height. Wind is typically already calculated correctly within less than 60 sec, respectively 1000 m".

Lines 231ff: The heating feature of the DFM-17 reduces the risk for evaporative cooling after exiting clouds. The lack of this feature in the dropsonde variant is relevant and could extend the perceived cloud thickness to lower altitudes and could potentially lead to evaporative cooling at cloud base. This potential limitation needs to be discussed. Does the sonde measure and report the temperature of the humidity sensor?

The dropsonde measures and reports the temperature of the humidity sensor. As the temperature sensor cannot be collocated on the humidity sensor, there is a small space between the two sensors, which leads to a larger uncertainty component than with the heated sensor. Nevertheless, the temperature data can be used to correct the humidity measurement for the temperature of the humidity sensor. We have rephrased the sentence to highlight this better. This setup is also able to detect situations, where the humidity sensor might experience more wet-bulbing than the ambient temperature sensor. The main difference between the two sensors for cloud-exits is the faster response time of the heated sensor. However, in the dropsonde application, response time decreases during flight progression, inverse to radiosonde operation.

The whole text block below Table 1 has been slightly rearranged. Especially:

Modified text (now lines 240-247): "The humidity sensor uses the same type of polymer but is unheated and has its accompanying temperature sensor located a few mm apart. The heated humidity sensor of the latest DFM-17 generation was not yet available at the time when the EL-18 was designed. The close temperature sensor can be used to correct the humidity measurement for the humidity sensor temperature. Still, this slightly reduces dynamic performance and response time, especially after the sensor has experienced precipitation. However, as the sensor response time decreases during the sounding, inverse to radiosonde operation, this problem is not as severe as with radiosoundings. As for most radiosonde algorithms (Dirksen et al. 2024), an active

clipping of measured relative humidity values > 100 %RH is performed. Humidity values before clipping of up to 120% could be observed, which is a typical value."

Do you have access to raw humidity data without clipping? That would provide some feeling about the issues of measuring RH near saturation. This should be discussed to

There is access to raw humidity values without clipping, and the humidity shows spikes in saturation conditions of up to 120 %, which is a typical value when small amounts of water are present on the sensor. We added a sentence to clarify this.

Added text in lines 245-247: "As for most radiosonde algorithms (Dirksen et al. 2024), an active clipping of measured relative humidity values > 100 %RH is performed. Humidity values before clipping of up to 120% could be observed, which is a typical value."

Section 2.4: This section should include the process how the meteorological sonde is released from its release container when satellite communication is used to transmit data to the ground directly. This is mentioned later in the manuscript.

The launching process in satellite configuration is in principal similar to the standard configuration with meteorological sondes. The release container detects the dropping process and activates the main parachute. After deceleration and vertical orientation of the release container the payload is set free. In this case the payload consists of (i) one meteorological sonde, connected via a cable of 5m length to (ii) the insert containing the satellite modem with communication antenna and batteries and (iii) the parachute. The size of the parachute is larger than those used for the small meteorological sondes to achieve decent rates of about 10 m/s.

The following text is added in lines 282-287: "The launching process in satellite configuration is similar to the standard configuration with meteorological sondes. The release container detects the dropping process and activates the main parachute. After deceleration and vertical orientation of the release container the payload is set free. In this case the payload consists of (i) one meteorological sonde, connected via a cable of 5m length to (ii) the insert containing the satellite modem with communication antenna and batteries and (iii) the parachute. The size of the parachute is chosen larger than those used for the small meteorological sondes to achieve decent rates in the order of, e.g., about 10 m s$^{-1}$."

Section 2.5: How can you be sure that the airflow through the particle detector is close to the ambient fall rate? Is the central channel straight without any obstructions? Can you account for any pendulum motion of the sonde, which would invariably affect the air flow through the instrument? Some discussion follows later but should be moved here. The meteorological sonde hanging below the OPC carries the risk of particle shedding, in particular after passing through clouds, which the particle detector may sense. This is probably not an issue for aerosol particles but should be mentioned.

Detailed descriptions of the particle detector (OPC) and its performance are provided in the references already cited (Smith et al., 2018; Girdwood et al., 2020), and answers to the Reviewer's questions can be found there. In particular, the channel is free of obstructions. As for the „pendulum motion", Smith et al. (2018) state: „*To constrict the movement of the UCASS, the payload is configured as a double pendulum...*", with the text which follows explaining that this configuration increases energy dissipation and restricts the motion of the OPC (which for both the balloon and the dropsonde case resides in the middle of the „train", as stated in the text). There are also descriptions of

both computational fluid dynamics modelling and flow tests, including the impact of flow direction angle, but too extensive to repeat in the present submission. We refer the Reviewer and the future readers to these publications. As for particle shedding, if it occurs, there has been no evidence that it affects the OPC, e.g. following passage through clouds (where number concentrations are more than an order of magnitude greater than outside – see Fig. 11).

Following the reviewer, we moved the section „After release, the meteorological sonde and the OPC separated, but remained tied together by a 1 m long polyester line, with the meteorological sonde located lowest, followed by the OPC, and the parachute attached at the top of the "train" (Fig. 6d). This configuration creates a double pendulum which tends to dissipate kinetic energy and helps to reduce swinging motions of the OPC, thus aligning it with the air velocity vector for correct particle sampling (Smith et al., 2019)." to this place (line 335 in revised paper).

Incidentally, the submitted text requires a correction in three places, Smith et al. is cited variously as 2018 and 2019, but only the 2019 year is correct.

Line 253: The statement about size and cost is relative and should be deleted here.

Accepted, i.e. the statement dropped in the manuscript

Section 2.6: Can you show the profile of the test drop near Magdeburg? There is possibly a detectable increase in radioactivity due to cosmic ray activity as a function of altitude, which would be worthwhile showing. This would give a feeling of the baseline that the detector would see.

We agree that such an increase in the detected signal would be a good indicator of the baseline the counter sees. Geiger-Müller-Counters, however, are known to be not ideal for cosmic ray measurements, since they lack energy discrimination and have low efficiency for detecting high-energy muons. As we do not see a clear increase in this single profile, we prefer not to show it and to gather more information with the system, before a discussion of quantitative data becomes possible.

For temperature, humidity, pressure, and wind it would be good to know what the true uncertainties in flight are after all systematic and random errors are considered. The uncertainties listed in Table 1 are extremely optimistic and unrealistic for real world observations. This becomes relevant when the reader tries to interpret the differences of soundings discussed in section 3. Large scale structures shown in Figure 9 are certainly real, but when the differences reach those of the stated uncertainties, caution is warranted.

The uncertainties stated in Table 1 are derived from the manufacturer's specifications, based on industrial sensor production processes and testing. Real world uncertainties depend on several changing influences, such as solar heating, and riming on sensors. There have been previous validation flights against radiosondes, but from these, no general quantitative uncertainties can be derived. We have added a paragraph in section 2.3 where we set the expected uncertainties in context to the uncertainties observed in the UAII 2022 campaign. This extense radiosonde intercomparison, although not fully applicable, gives important information on the DFM-17, the EL-18 being derived from it's electronic circuity and sensors (with some changes, discussed in the proposed added text below). The authors are not aware of sufficient experimental approach for

dropsondes, neither by other manufacturers or at all, to assess similarly the real world uncertainties.

Additional text in lines 230-244:

"The uncertainties stated in Table 1 are derived from the manufacturer's specifications. To estimate whether these can be used as uncertainty estimates for the EL-18 dropsonde electronics, a closer look at the WMO intercomparison campaign UAII 2022 (Dirksen et al., 2024) is undertaken. In particular, the sensors and analogue to digital converter circuity as well as the manufacturing and calibration processes are the same for both sondes. The EL-18 uses a slightly modified sensor boom shape to accommodate the dropsonde application (Fig. 5b). Due to the different nature of the applications, not all corrections used in the DFM-17 data product can be used for the EL-18. The measurement method for all GNSS-derived variables, such as wind and geopotential height, all measurements are identical between the two sondes. For pressure, there might be a small decrease in uncertainty near the ground due to the dedicated barometric pressure sensor, compared to the DFM-17 used in the UAII 2022 campaign. The temperature sensor is also identical between the two models, however, the dropsonde uses a slightly altered solar correction model to account for the different orientation of the sensor boom and the adjacent parts of the housing. The humidity sensor uses the same type of polymer but is unheated and has its accompanying temperature sensor located a few mm apart.  The heated humidity sensor of the latest DFM-17 generation was not yet available at the time when the EL-18 was designed. The close temperature sensor can be used to correct the humidity measurement for the humidity sensor temperature. Still, this slightly reduces dynamic performance and response time, especially after the sensor has experienced precipitation."

A particular challenge for dropsonde measurements is their validation. Are there any baseline measurements on the aircraft prior to launch? Are there any additional measurements on the ground prior to a mission? Have there been any attempts to compare the dropsonde observations with nearby balloon or remote sensing (e.g. lidar) measurements? Have there been any attempts to have four sondes from a release container to measure as close by as possible, i.e. using identical parachute size? If so, it would be good to show the scatter of all four sondes around their mean. Any additional information that could be used to quantify the accuracy of the measurements in flight would be helpful to get a better impression about the true capabilities of the meteorological sondes.

The general part of the question is similar to the one above, and is answered there. In addition:

- no ground measurements are done before flights, other to radiosondes; ground data before a flight would also differ much to those of dropsondes landed later and elsewhere
- aircraft data from flight level are available for rough comparison with dropsondes once adapted to the surroundings (i.e. a few hundred meters after dropping)
- comparisons between several sondes with identical parachutes have not yet been done; several hundred of sondes launched during the upcoming ASCCII and NAWDIC campaigns will provide opportunities. But we optimistically expect, that sondes following industrial production standards will under same conditions behave according to the given uncertainties.

Line 379f: You correctly state that the atmosphere above 2 km is stably stratified. What then does the statement refer to that says "Two more stable shallow layers are present at about 6 and 8 km AMSL", when the entire region is stable? Please clarify.

We rewrote the sentences for clarification (new lines 444-447)

"The four temperature profiles in the area east of the Sierras de Córdoba show a similar vertical stratification (Fig. 10a). A well-mixed boundary layer reaches up to approximately 2 km AMSL and stable stratification of approximately 4 K km$^{-1}$ is found above. Additionally, in the data of sonde 1a and 1b shallow layers with stronger stability can be found at the top of the cloud layer, i.e. between about 5.5 and 6.0 km AMSL (Fig. 10b)."

Line 381f: There seems to be a problem with the color coding of the profiles or with referencing the profiles in the text. The cloud top in profile 1d seems to be closer to 4.5 km (not 5.5 km). Profile 1c (not 1b) does not appear to fall through a cloud. Please check the description of the text of this section.

The suggested modifications were applied (new lines 447-451)

"The largest differences between the profiles of the sondes are obtained for relative humidity (Fig. 10b). While sonde 1d measures the cloud top at about 4.5 km AMSL already, sonde 1c did not even fall into a cloud at all. The cloud bases also vary considerably, i.e. between about 1.5 km (sonde 1b) and about 3 km AMSL (sondes 1a and 1d). Thus, the high thermodynamic variability in areas with deep convection is captured well by this kind of quadruple information – which cannot be provided by individual dropsondes or single radiosondes."

Is there a reason that all four sondes of drop 1 seem to be missing data in the region between 10 km and 11 km? The wind speed in that layer seems to be at or above 50 m/s (not 40 m/s as stated).

The missing data between 10 and 11 km are due to transmission problems for unknown reasons, i.e. no data could be received from the sondes. 40 m/s has been replaced by 50 m/s (also indicated in red in the text)

"The horizontal wind speed varies between 5 m s$^{-1}$ at 2 km AMSL and 50 m s$^{-1}$ at 10 km AMSL (Fig. 10c) in all profiles."

The new textblock in the revised version (lines 444-454-388 with changes in red) is:

"The four temperature profiles in the area east of the Sierras de Córdoba show a similar vertical stratification (Fig. 10a). A well-mixed boundary layer reaches up to approximately 2 km AMSL and stable stratification of approximately 4 K km$^{-1}$ is found above. Additionally, in the data of sonde 1a and 1b shallow layers with stronger stability can be found at the top of the cloud layer, i.e. between about 5.5 and 6.0 km AMSL (Fig. 10b). The largest differences between the profiles of the sondes are obtained for relative humidity (Fig. 10b). While sonde 1d measures the cloud top at about 4.5 km AMSL already, sonde 1c did not even fall into a cloud at all. The cloud bases also vary considerably, i.e. between about 1.5 km (sonde 1b) and about 3 km AMSL (sondes 1a and 1d). Thus, the high thermodynamic variability in areas with deep convection is captured well by this kind of quadruple information – which cannot be provided by individual dropsondes or single radiosondes. The horizontal wind speed varies between 5 m s$^{-1}$ at 2 km AMSL and 50 m s$^{-1}$ at 10 km

AMSL (Fig. 10c) in all profiles. The wind direction in the boundary layer shows some veering from northerly to westerly winds. It is predominantly northwesterly between 2 and 6 km AMSL and westerly above (Fig. 10d). This vertical profile of the horizontal wind is visible in all four soundings. "

Lines 414ff: This explanation about the OPC sounding should be moved up to the instrument description in section 2.5.

Yes, we agree, that fits better.

The sentences

"After release, the meteorological sonde and the OPC separate, but remain tied together by a 1 m long polyester line, with the meteorological sonde located lowest, followed by the OPC, and the parachute attached at the top of the "train" (Fig. 6d). This configuration creates a double pendulum which tends to dissipate kinetic energy and helps to reduce swinging motions of the OPC, thus aligning it with the air velocity vector for correct particle sampling (Smith et al., 2019)."

are moved to section 2.5.

Line 424: Figures 13 and 14 are referred to before Figures 11 and 12. Please add a reference to Figure 11 and 12 before that.

Checked and modified

Line 472f: Paragliders require active control or the parafoil. Are there any plans to implement this? Otherwise, maybe delete this statement.

We think of parafoils. As a future option this seems worth mentioning, since there are several inexpensive  models available. The container concept of the KITsonde allows for simple ways of testing.

Paragliders replaced by parafoils.

Lines 494ff: Although I don't doubt this capability, you only showed evidence for one Sahara dust layer, but not for any aerosol layer embedded in clouds. If such results exist, it would be good to include them, otherwise this statement should be revised.

The sentence in 494ff reads: "Tests presented here demonstrate that the soundings do not only yield the profiles of coarse aerosol, but also allow for the identification and characterisation of embedded cloud layers." We do not claim to be able to identify aerosol layers within clouds.
While cloud layers embedded in dust are easy to identify in UCASS data, the reverse is not necessarily the case. This is because number concentrations of larger particles are many times greater in clouds than in dust layers. Consequently, in clouds the presence of coarse dust is „masked" by the presence of droplets – see Fig. 11. Having said that, the presence of dust within a cloud could be apparent if the OPC was calibrated to count particles within a lower, submicron size range, below the size range for cloud droplets. However, while technically possible, this remain hypothetical.

We believe the text is correct as stated

Line 507f: What is meant by "aviation consulting processes"? Please clarify.

Line 508f: Similarly, what is meant by "a concept for validation based on measurement data"?

We clarified the sentences in Line 577 and 581 and modified the description (new lines 569-573).

"One aim of MEASURE is the implementation of dropsondes into a drone-based measurement system for use in a radiological emergency. Quickly available dropsonde profiles can offer substantial added value in evaluating potential pollutant distributions in the airspace, thus supporting the aviation consulting process of DWD. In addition, real-time data from dropsondes provide a valuable contribution to the validation of radiological dispersion calculations of DWD, which serve as a reliable information basis for decision-makers."

Line 62: The reference by Hartmann et al., 1996 does not mention any dropsonde releases. The Berichte zur Polarforschung Nr. 218 was published in 1997. If dropsondes were released during that project, a different reference would be helpful.

That is right, we added a reviewed paper (Chechin et al., 2013) instead, that makes extended use of these dropsondes.

Chechin, D. G., Lüpkes, C., Repina, I. A., & Gryanik, V. M.: Idealized dry quasi 2-Dmesoscale simulations of cold-air outbreaks over the marginal sea ice zone with fine and coarse resolution. J. Geophys. Res. Atmos., 118(16), 8787-8813, https://doi.org/10.1002/jgrd.50679, 2013.

**Table 1:**

Table 1 lists uncertainties and reproducibilities. What is their significance level, i.e. one standard deviation (k=1) or two standard deviations (k=2) or something else?

The significance level is given as k=2. We have added a footnote to the table to clarify.

Footnote Table 1: Uncertainties are expressed with 2-sigma confidence level (k=2).

The datasheet for the Bosch Sensortec pressure sensor lists its operating range as 300... 1250 hPa. Where does the measurement range to 1 hPa and the uncertainties at the lower pressures come from?

The measurement data from the BMP388 is used at lower altitudes, while a GNSS derived pressure is used at higher altitudes. This approach combines the advantages of both barometric pressure sensors and GNSS-derived pressures. We have added a footnote to the table to clarify.

Footnote Table 1: GNSS derived pressure is used to augment the barometric pressure measurement at lower pressures.

The measurement range for the wind speed is probably larger than 200 m/s and the sonde can probably sense the aircraft speed prior to launch.

Thank you for pointing that out, the maximum value is indeed 500 m/s. Yes, GPS reception is only interrupted in the dispenser tube and 100 – 200 m below the aircraft.

We have corrected the value in the table accordingly.

**Figures:**

Figure 5: Where is the humidity sensor located and what is the airflow around it?

We replaced Fig. 5b by a figure showing the location of the humidity and the temperature sensor better.

Figure 8, legend:  change "white circles" to "white dots"

Right, done.

Figure 11, legend: Please make sure that the unit µm is displayed properly (also in lines 438 and 440). Capitalize "Individual".

That has been corrected.

**Technical comments:**

Lines 33-35: Rephrase to improve the clarity of that statement.

Corrected text (changes in red)

Moreover, a configuration consisting of a meteorological sonde coupled with an optical counter for particle sizing was tested during a Saharan dust episode over Germany using a Dornier Do 128-6 aircraft.  Secondly, a meteorological sonde together with a radioactivity sensor was successfully dropped from a Learjet 35A.

Line 48: A better reference to the NCAR dropsonde is:

The reference is replaced as proposed.

Hock, T. F., and J. L. Franklin, 1999: The NCAR GPS Dropwindsonde. Bull. Amer. Meteor. Soc., 80, 407–420, https://doi.org/10.1175/1520-0477(1999)080<0407:TNGD>2.0.CO;2.

 "Airborne" replaces "Advanced"

"on" changed to "for"

Line 145: insert a hyphen into "system-control"

Done

Line 189: replace "independently" with ""independent"

Done

Line 319: Add "/" to "Upper Troposphere/Lower Stratosphere"

Done

Line 338: The distance given sounds a little short.

Due to strong headwinds the airspeed above ground was less.

We replaced by

At true air speed of 200 m/s and strong head winds of 50 m/s, 1 to 2 min corresponds 9 to 18 km.

Line 339: What are typical dropsonde parachute cross sections?

A new section on the parachutes for the EL-18 and their dimensions is added in Section 2. We analysed the fall speeds depending on effective size in dependence on height. (see below)

The parachutes for the container, for the coupled radioactivity/meteorological measurements, and the particle/meteorological measurements are different and of the shape of the figure below. It consists of 5 quadratic sections, each 15 x 15 cm. The total area is 1125 cm$^2$.

[Figure]

New section added:

**2.3.2 Parachutes**

The parachutes are chosen to determine the descent rates, which affects how closely the sounding trajectory approximates a vertical profile as well as how closely the sounding approximates to an "instantaneous" profile. The parachutes of 64 cm in diameter for the EL-18 being used are manufactured by BBL Elektronik & Aeromet GmbH (https://www.meteorologyshop.eu/en/balloons/radiosonde-balloons/368/meteorological-parachute-pc-055). To allow for different fall speeds, the effective areas of the parachutes were reduced by cutting away outer trapezoidal sectors of fabric between the holding lines. Tab. 2 shows the effective diameters, corresponding to the effective area of the parachute. Fall speeds versus parachute size were analysed for the SouthTRAC Campaign (see Sect. 3), where 60 meteorological sondes EL-18 without any additional sensor or communication electronics were used.

**Table 2: Mean fall speed and standard deviations in m s$^{-1}$ of meteorological sondes dropped with parachutes of different effective diameters. The fall speeds are given for different height ranges during the SouthTRAC campaign.**

| Height range (km) | Parachute effective diameter (cm) in line 2 and area (cm$^2$) in line 3 for the meteorological sonde EL-18 of 73 g weight | | | |
|---|---|---|---|---|
| | 64 | 40 | 32 | 24 |
| | 3217 | 1257 | 804 | 452 |
| 8 - 12 | 4,5±1,2 | 7,0±2,1 | 8,1±2,1 | 11,2±1,6 |
| 4 - 8 | 3,3±1,0 | 4,9±1,5 | 5,9±1,4 | 8,7±1,0 |
| 0 - 4 | 2,6±0,6 | 3,8±1,2 | 4,8±1,1 | 7,3±0,5 |

The parachutes for the container, for the coupled radioactivity/meteorological measurements and for the particle/meteorological measurements are more robust and of a x-pentamine shape. They are manufactured by Spekon Co. (https://spekon.de/seilschirme.html) and consists of 5 quadratic sections, each 15 x 15 cm. The total area is 1125 cm$^2$.

Line 466: Replace "aöö" with "all"

Done

Lines 355 and 470: Please specify the altitude for this fall rate. Or alternatively, specify the fall rate near the surface.

A new section on the parachutes is added in Section 2

Line 484: Replace "exposition" with "exposure"

done

Table 1 lists a Tellit J-N3 GNSS module as well as a UBLOX MAX-MBC. I assume the text is correct and that the UBLOX MAX-M8C is used. Please correct Table 1.

The module correctly is „u-blox MAX-M8C"

Corrected in Table 1

---

## Author Response (AR2)

**Reply to the Reviewers comments on the revised manuscript**

One reviewer had no more comments on the revised manuscript-

The second reviewer had four smaller comments (corrections) on the revised manuscript. These corrections were accepted and the final manuscript was changed accordingly.